# Dietary circadian rhythms and cardiovascular disease risk in the prospective NutriNet-Santé cohort

Anna Palomar-Cros [1,2], Valentina A. Andreeva[3], Léopold K. Fezeu[3], Chantal Julia[3,4], Alice Bellicha[3,5], Emmanuelle Kesse-Guyot[3,5], Serge Hercberg[3,4,5], Dora Romaguera[1,6,7], Manolis Kogevinas[1,2,8,9], Mathilde Touvier [3,5,10] & Bernard Srour [3,5,10] ✉

Daily eating/fasting cycles synchronise circadian peripheral clocks, involved in the regulation of the cardiovascular system. However, the associations of daily meal and fasting timing with cardiovascular disease (CVD) incidence remain unclear. We used data from 103,389 adults in the NutriNet-Santé study. Meal timing and number of eating occasions were estimated from repeated 24 h dietary records. We built multivariable Cox proportional-hazards models to examine their association with the risk of CVD, coronary heart disease and cerebrovascular disease. In this study, having a later first meal (later than 9AM compared to earlier than 8AM) and last meal of the day (later than 9PM compared to earlier than 8PM) was associated with a higher risk of cardiovascular outcomes, especially among women. Our results suggest a potential benefit of adopting earlier eating timing patterns, and coupling a longer nighttime fasting period with an early last meal, rather than breakfast skipping, in CVD prevention.

Cardiovascular diseases (CVD) are the leading cause of mortality and disease burden in the world[1]. Diet is a major risk factor for cardiovascular disease and contributes to 7.94 million CVD-related deaths annually[1]. The accelerated modern lifestyle linked with the perception of lacking time in Western societies, and the current surge in fasting practices promoting meal skipping, has led to mistimed nutritional behaviours, such as late-night eating and breakfast skipping[2]. The daily eating/fasting cycle is a dominant synchroniser of circadian rhythms in peripheral organs[3] including mainly the liver, but also the heart, kidney and pancreas, and has an influence on cardiometabolic functions including the regulation of blood pressure[4–6]. Chrononutrition has

emerged as a new field in nutritional sciences to unravel the relationship between timing of food intake, circadian rhythms and health[7].

Data from observational and interventional studies indicate that breakfast consumption is an important habit for cardiometabolic health[8] while its omission has been associated in meta-analyses with overweight and obesity[9], risk of CVD[10] and diabetes mellitus[11]. Similarly, late-night eating has been linked in prospective studies to cardiovascular risk factors such as arterial stiffness[12], obesity, dyslipidaemia, metabolic syndrome (in women only)[13] and to a higher risk of coronary heart disease in one prospective study[14]. However, as stated by the American Heart Association (AHA), an important

[1]Barcelona Institute for Global Health (ISGlobal), 08003 Barcelona, Spain. [2]Department of Experimental and Health Sciences, Universitat Pompeu Fabra (UPF), 08003 Barcelona, Spain. [3]Université Sorbonne Paris Nord and Université Paris Cité, INSERM, INRAE, CNAM, Center of Research in Epidemiology and StatisticS (CRESS), Nutritional Epidemiology Research Team (EREN), Bobigny, France. [4]Public Health Department, Avicenne Hospital, AP-HP, Bobigny, France. [5]Nutrition And Cancer Research Network (NACRe Network), Jouy-en-Josas, France. [6]Health Research Institute of the Balearic Islands (IdISBa), 07120 Palma de Mallorca, Spain. [7]CIBER Fisiopatología de la Obesidad y Nutrición (CIBEROBN), 28029 Madrid, Spain. [8]Hospital del Mar Medical Research Institute (IMIM), 08003 Barcelona, Spain. [9]Consortium for Biomedical Research in Epidemiology and Public Health (CIBERESP), Institute of Health Carlos III, 28029 Madrid, Spain. [10]These authors contributed equally: Mathilde Touvier, Bernard Srour. ✉e-mail: b.srour@eren.smbh.univ-paris13.fr

limitation of these studies is the lack of consensus in defining a meal[8]. Definitions are usually based on the participant identification of a meal (i.e. breakfast) and on the time of day (i.e. considering breakfast anything between 6 to 10AM). These approaches are prone to classification bias (i.e. eating at 11AM could be considered for one person as having breakfast but not for another person; the interpretation of late-eating could also be misleading depending on the cultural differences). Only two studies of cross-sectional design have evaluated specifically meal timing, in its continuous form, in relation with cardiovascular risk factors (intermediate endpoints) but none with CVD risk (hard endpoints)[15,16].

On the other hand, a growing body of evidence suggests that practicing time-restricted eating (TRE) (i.e., extending the nighttime fasting duration to more than 12 hours) could be linked to an improvement of multiple key indicators of cardiovascular health. In animal models, restricting feeding only to their active phase (to an 8- or 10-h window) has been associated with improved metabolic health and protection against obesity[17,18]. In humans, TRE has also been linked with reduced body weight, blood pressure and inflammation[19–22]. However, results have not always been consistent, since other studies have reported no association with cardiometabolic risk factors[23–25]. To our knowledge, no study has so far investigated the direct association between nighttime fasting duration and CVD risk (hard endpoint). Moreover, as stated by the AHA[26], epidemiological data on number of eating occasions and cardiometabolic risk are scarce and inconclusive and further studies should evaluate this behaviour in the context of meal timing and nighttime fasting duration.

The main objective of the present study is to explore the associations of time of first and last meal of the day, number of eating occasions and nighttime fasting duration, with the risk of CVD, in the prospective NutriNet-Santé cohort. These new findings suggest that adopting earlier daily eating patterns may be beneficial for cardiovascular prevention

## Results

The present study included a total of 103,389 participants (79% women) with a mean baseline age of 42.6 years (14.5 SD). Participants completed on average 5.7 dietary records (SD 3.0) with a maximum of 15 records. The main characteristics of the study population and according to their meal timings are shown in Table 1. Overall, younger participants, students or unemployed, single, without a family history of CVD, current regular smokers, with higher physical activity levels, higher educational levels and lower monthly incomes tended to have later first and last meals. Additionally, compared to participants with earlier meals, participants having later meals had a higher consumption of alcohol, more episodes of binge drinking, reported later bedtimes and were more likely to have a higher variability in their meal timings across the week (defined as eating jet lag). The distribution of meal timings and frequency are shown in Supplementary Figure S3.

### Association of meal timing and number of eating occasions with CVD risk

During a median follow-up time of 7.2 years (1st quartile [Q1] – 3rd quartile [Q3], 3.1–10.1) and 699,547 person-years, 2036 incident cases of CVD were ascertained. There were 988 cases of cerebrovascular diseases (253 cases of stroke and 765 of transient ischemic attack) and 1071 cases of coronary heart diseases (162 cases of myocardial infarction, 428 of angioplasty, 89 of acute coronary syndrome and 428 of angina pectoris). Our primary Cox models examining the association of meal timing and number of eating occasions with risk of overall CVD, cerebrovascular diseases and coronary heart diseases are shown in Table 2. The proportional hazard assumption was met, and there was no evidence for non-linear associations (Figs. S4 and S5). There was no evidence of collinearity among covariates.

We observed that each additional hour in delaying the time of first meal of the day was associated with a higher risk of overall CVD (Table 2, HR = 1.06, 95% CI 1.01–1.12, P-value = 0.02). The continuous variable of the time of last meal was not significantly associated with CVD risk. The HR linking time of the last meal with overall CVD risk was 1.13 (0.99–1.29), P-trend = 0.06, for having a last meal after 9PM compared to before 8PM. We observed no association with the number of eating occasions.

In this study population, time of first meal of the day was not associated with the risk of cerebrovascular disease (Table 2). However, each additional hour in delaying the time of last meal was associated with an 8% increased risk of cerebrovascular disease (HR = 1.08, 95% CI 1.01–1.15, P-value = 0.02): more specifically, compared to a last meal before 8PM, a last meal after 9PM was associated with a 28% higher risk of cerebrovascular disease (HR = 1.28, 95% CI 1.05–1.55, P-trend < 0.01). No association was observed for a number of eating occasions.

We found no association between meal timing nor number of eating occasions and risk of coronary heart disease in this study (Table 2).

### Association of nighttime fasting duration with CVD risk

We found an inverse association between nighttime fasting duration and cerebrovascular disease risk (Table 3). Each additional hour of nighttime fasting was associated with a 7% lower risk of cerebrovascular disease (HR = 0.93, 95% CI, 0.87–0.99, P-value = 0.02), but not with risk of overall CVD or coronary heart disease. We did not find an interaction between nighttime fasting duration and time of first meal for the associations with overall CVD, cerebrovascular disease nor coronary heart disease (p-values for interaction = 0.2, 0.3, 0.5, respectively).

### Differences by sex

We found a statistically significant interaction between sex and time of last meal of the day for the associations with overall CVD (p-value = 0.01) and coronary heart disease risk (p-value = 0.004). In Table 4 we present the results of the primary models considering the interaction of sex with each of the exposure variables. Globally, our results suggest stronger associations in women than in men. Specifically, our results show that later times of first and last meals were significantly associated with a higher risk of overall CVD and cerebrovascular disease for women but not for men (Table 4). We also observed a significant interaction between sex and nighttime fasting duration for the association with coronary heart disease (p-value = 0.02). The results for the number of eating occasions were null and similar for both sexes.

### Sensitivity analyses

The associations remained relatively stable across all sensitivity analyses (Table S1). The association between the time of first meal of the day and the risk of overall CVD was moderately attenuated after considering chronotype in the sample restricted to participants with data on sleep duration (Table S1, model 10). As a falsification endpoint, neither the time of first nor last meal of the day were associated with basal cell carcinoma risk in this population (Table S1, Model 19). Although in the same direction as in the main model, the link between the time of last meal and risk of cerebrovascular disease was attenuated and no longer significant in the reduced sample with participants responding to the sleep questionnaire (Table S1, Model 7), probably due to a substantial loss of statistical power. Adjusting for sleep quality indicators (Model 11), marital status and number of children (Model 13) did not change the observed associations. Similarly, adding the season of completion of the physical activity questionnaire and the first set of dietary records did not alter our results (Model 15 and 18). The inclusion of a number of medications (Model 14), unusual dietary reporting (Model 16) and restrictive diets (Model 17) did not

**Table 1 | Baseline characteristics of the included participants from the NutriNet-Santé cohort, 2009–2021, N = 103,389**

| | All participants | Time of first meal | | | Time of last meal | | |
|---|---|---|---|---|---|---|---|
| | N = 103,389 N (%) or mean (SD) | Before 8AM N = 46,306 N (%) or mean (SD) | 8AM to 9AM N = 36,981 N (%) or mean (SD) | After 9AM N = 20,102 N (%) or mean (SD) | Before 8PM N = 34,723 N (%) or mean (SD) | 8PM to 9PM N = 45,610 N (%) or mean (SD) | After 9PM N = 23,056 N (%) or mean (SD) |
| Age at baseline | 42.6 (14.5) | 46.5 (13.1) | 42.5 (14.8) | 33.9 (13.3) | 45.7 (14.7) | 41.7 (14.1) | 39.8 (14.1) |
| Sex | | | | | | | |
| Women | 81,709 (79.0) | 35,699 (77.1) | 29,817 (80.6) | 16,193 (80.6) | 27,284 (78.6) | 36,826 (80.7) | 17,599 (76.3) |
| Men | 21,680 (21.0) | 10,607 (22.9) | 7164 (19.4) | 3909 (19.4) | 7439 (21.4) | 8784 (19.3) | 5457 (23.7) |
| BMI (kg/m$^2$) | 23.8 (4.5) | 23.9 (4.43) | 23.7 (4.42) | 23.5 (4.74) | 24.0 (4.49) | 23.6 (4.41) | 23.7 (4.62) |
| Family history of CVD | | | | | | | |
| No | 70,649 (69.4) | 30,447 (65.8) | 25,804 (69.8) | 15,967 (79.4) | 23,272 (67.0) | 32,277 (70.8) | 16,669 (72.3) |
| Yes | 31,171 (30.6) | 15,859 (34.2) | 11,177 (30.2) | 4135 (20.6) | 11,451 (33.0) | 13,333 (29.2) | 6387 (27.7) |
| Smoking status[a] | | | | | | | |
| Regular current | 12,442 (12.0) | 4621 (9.99) | 4184 (11.3) | 3637 (18.1) | 2942 (8.48) | 5144 (11.3) | 4356 (18.9) |
| Occasional current | 5425 (5.25) | 1772 (3.83) | 2069 (5.60) | 1584 (7.89) | 1358 (3.91) | 2476 (5.43) | 1591 (6.91) |
| Former | 33,563 (32.5) | 16,833 (36.4) | 11,941 (32.3) | 4789 (23.9) | 11,856 (34.2) | 14,859 (32.6) | 6848 (29.7) |
| Never | 51,866 (50.2) | 23,040 (49.8) | 18,760 (50.8) | 10,066 (50.1) | 18,537 (53.4) | 23,090 (50.7) | 10,239 (44.5) |
| Packs per year[b] | 5.30 (11.3) | 6.01 (12.1) | 5.04 (10.9) | 4.18 (9.84) | 5.20 (11.6) | 4.92 (10.5) | 6.20 (12.3) |
| Physical activity[c] | | | | | | | |
| Low | 29,164 (32.7) | 14,762 (31.9) | 9876 (26.7) | 4526 (22.5) | 10,470 (30.2) | 12,341 (27.1) | 6353 (27.6) |
| Moderate | 21,674 (24.3) | 22,637 (48.9) | 19,224 (52.0) | 10,690 (53.2) | 17,155 (49.4) | 23,573 (51.7) | 11,823 (51.3) |
| High | 38,334 (43.0) | 8907 (19.2) | 7881 (21.3) | 4886 (24.3) | 7098 (20.4) | 9696 (21.3) | 4880 (21.2) |
| Daily alcohol intake[d] | | | | | | | |
| Non-consumers | 32,102 (31.0) | 13,837 (29.9) | 10,624 (28.7) | 7641 (38.0) | 12,432 (35.8) | 12,975 (28.4) | 6695 (29.0) |
| Low consumption | 26,897 (26.0) | 12,670 (27.4) | 9840 (26.6) | 4387 (21.8) | 9419 (27.1) | 12,155 (26.6) | 5323 (23.1) |
| Medium consumption | 26,241 (25.4) | 11,997 (25.9) | 9812 (26.5) | 4432 (22.0) | 8023 (23.1) | 12,311 (27.0) | 5907 (25.6) |
| High consumption | 12,658 (12.2) | 5545 (12.0) | 4718 (12.8) | 2395 (11.9) | 3522 (10.1) | 5811 (12.7) | 3325 (14.4) |
| Very high consumption | 5491 (5.31) | 2257 (4.87) | 1987 (5.37) | 1247 (6.20) | 1327 (3.82) | 2358 (5.17) | 1806 (7.83) |
| Episodes of binge drinking[e] | | | | | | | |
| None | 92,700 (89.7) | 42,301 (91.4) | 32,959 (89.1) | 17,440 (86.8) | 32,183 (92.7) | 40,587 (89.0) | 19,930 (86.5) |
| One | 7888 (7.6) | 2971 (6.4) | 2985 (8.1) | 1932 (9.6) | 1943 (5.6) | 3722 (8.2) | 2223 (9.6) |
| More than one | 2788 (2.7) | 1029 (2.2) | 1032 (2.8) | 727 (3.6) | 594 (1.7) | 1296 (2.8) | 898 (3.9) |
| Daily energy intake (kcal) | 1847 (451) | 1856 (448) | 1857 (441) | 1810 (475) | 1789 (434) | 1852 (437) | 1926 (490) |
| Higher education | | | | | | | |
| No | 17,868 (17.3) | 9476 (20.5) | 5851 (15.8) | 2541 (12.7) | 8042 (23.2) | 6669 (14.6) | 3157 (13.7) |
| Yes, <2 y after high-school | 16,318 (15.8) | 7166 (15.5) | 5451 (14.7) | 3701 (18.4) | 6054 (17.4) | 6918 (15.2) | 3346 (14.5) |
| Yes, ≥2 y after high-school | 69,133 (66.9) | 29,635 (64.0) | 25,659 (69.4) | 13,839 (68.9) | 20,604 (59.4) | 31,992 (70.2) | 16,537 (71.8) |
| Income per unit of consumption | | | | | | | |
| Less than 900€ | 11,381 (11.0) | 3550 (7.67) | 3829 (10.4) | 4002 (19.9) | 3619 (10.4) | 4680 (10.3) | 3082 (13.4) |
| 900–1200€ | 6761 (6.55) | 3060 (6.62) | 2342 (6.34) | 1359 (6.77) | 2478 (7.14) | 2870 (6.30) | 1413 (6.14) |
| 1200€ – 1800€ | 25,387 (24.6) | 11,547 (25.0) | 9030 (24.4) | 4810 (24.0) | 8975 (25.9) | 10,918 (24.0) | 5494 (23.9) |
| 1800€ – 2300€ | 14,721 (14.3) | 6929 (15.0) | 5228 (14.1) | 2564 (12.8) | 4953 (14.3) | 6495 (14.3) | 3273 (14.2) |
| 2300€ – 3700€ | 23,033 (22.3) | 11,247 (24.3) | 8573 (23.2) | 3213 (16.0) | 7459 (21.5) | 10,621 (23.3) | 4953 (21.5) |
| More than 3700€ | 9761 (9.45) | 4903 (10.6) | 3654 (9.89) | 1204 (6.00) | 2789 (8.04) | 4716 (10.4) | 2256 (9.80) |
| Don't wish to answer | 12,236 (11.8) | 5022 (10.9) | 4294 (11.6) | 2920 (14.5) | 4412 (12.7) | 5265 (11.6) | 2559 (11.1) |
| Profession | | | | | | | |
| Unemployed | 11,818 (11.5) | 4128 (8.94) | 4417 (12.0) | 3273 (16.4) | 4018 (11.6) | 4968 (10.9) | 2832 (12.3) |
| Student | 6821 (6.62) | 1320 (2.86) | 2231 (6.06) | 3270 (16.3) | 1877 (5.43) | 3085 (6.79) | 1859 (8.09) |
| Self-employed/farmer | 2649 (2.57) | 1329 (2.88) | 871 (2.36) | 449 (2.24) | 785 (2.27) | 1221 (2.69) | 643 (2.80) |
| Employed | 65,647 (63.7) | 31,096 (67.4) | 22,877 (62.1) | 11,674 (58.3) | 20,313 (58.7) | 30,123 (66.3) | 15,211 (66.2) |
| Retired | 16,077 (15.6) | 8285 (17.9) | 6447 (17.5) | 1345 (6.72) | 7598 (22.0) | 6055 (13.3) | 2424 (10.6) |
| Marital status | | | | | | | |
| Single | 17,516 (17.0) | 5540 (12.0) | 5876 (15.9) | 6100 (30.4) | 4981 (14.4) | 7348 (16.1) | 5187 (22.5) |
| Married or in couple | 74,888 (72.6) | 35,083 (76.0) | 27,276 (74.0) | 12,529 (62.5) | 25,647 (74.1) | 33,743 (74.2) | 15,498 (67.4) |
| Divorced, separated or widower | 10,723 (10.4) | 5566 (12.1) | 3730 (10.1) | 1427 (7.12) | 3995 (11.5) | 4410 (9.69) | 2318 (10.1) |

**Table 1 (continued) | Baseline characteristics of the included participants from the NutriNet-Santé cohort, 2009–2021, N = 103,389**

|  | All participants | Time of first meal | | | Time of last meal | | |
|---|---|---|---|---|---|---|---|
|  | N = 103,389<br>N (%) or<br>mean (SD) | Before 8AM<br>N = 46,306<br>N (%) or<br>mean (SD) | 8AM to 9AM<br>N = 36,981<br>N (%) or<br>mean (SD) | After 9AM<br>N = 20,102<br>N (%) or<br>mean (SD) | Before 8PM<br>N = 34,723<br>N (%) or<br>mean (SD) | 8PM to 9PM<br>N = 45,610<br>N (%) or<br>mean (SD) | After 9PM<br>N = 23,056<br>N (%) or<br>mean (SD) |
| Number of used medications at baseline | 1.26 (1.83) | 1.20 (1.77) | 1.31 (1.85) | 1.31 (1.91) | 1.28 (1.83) | 1.22 (1.75) | 1.32 (1.97) |
| Time of first meal (AM) | 8:14 (1.1) | 7:22 (0.48) | 8:24 (1.63) | 9:57 (1.02) | 7:56 (1.01) | 8:14 (1.02) | 8:42 (1.32) |
| Time of last meal (PM) | 8:24 (1.1) | 8:06 (1.02) | 8:14 (1.00) | 8:48 (1.28) | 7:18 (0.74) | 8:24 (0.28) | 9:48 (0.90) |
| Eating jet lag |  |  |  |  |  |  |  |
| Advance | 5667 (5.6) | 2108 (4.7) | 1669 (4.6) | 1890 (9.7) | 2308 (6.8) | 1311 (2.9) | 2048 (9.1) |
| Maintenance | 69,867 (69.3) | 33,421 (74.3) | 25,244 (69.5) | 11,202 (57.4) | 24,171 (71.3) | 32,299 (72.4) | 13,397 (59.9) |
| Delay | 25,322 (25.1) | 9469 (21) | 9420 (26) | 6433 (32.9) | 7417 (21.9) | 10,972 (24.6) | 6933 (31.0) |
| Nighttime fasting hours | 11.9 (1.4) | 11.2 (1.09) | 12.0 (1.01) | 13.2 (1.51) | 12.6 (1.26) | 11.8 (1.02) | 10.9 (1.46) |
| Number of eating occasions/day | 4.89 (1.7) | 5.00 (1.78) | 4.89 (1.63) | 4.63 (1.71) | 4.41 (1.40) | 4.82 (1.55) | 5.73 (2.14) |
| Bedtime[f] | 23:42 (0.94) | 23:30 (0.84) | 23:54 (0.86) | 24:12 (1.15) | 23:24 (0.88) | 23:48 (0.84) | 24:12 (1.05) |
| Sleep duration (h) | 6.93 (1.59) | 6.75 (1.56) | 7.03 (1.60) | 7.24 (1.59) | 6.94 (1.65) | 6.95 (1.57) | 6.86 (1.53) |

*BMI* body mass index, *CVD* cardiovascular diseases, *N* sample size, *SD* standard deviation.
[a]Regular current smokers were participants smoking 1 or more cigarettes per day.
[b]The variable packs years smoking was calculated as the packs smoked per day per the years smoking.
[c]Physical activity was categorised according to International Physical Activity Questionnaire (IPAQ) guidelines.
[d]Daily alcohol intake was categorised as follows: Non-consumers (0 g/day), low consumers (0.1–4.9 g/day), moderate consumers (5.0–14.9 g/day), high consumers (15.0–29.9 g/day) and very high consumers (>30.0 g/day).
[e]An episode of binge drinking was considered for men if they consumed 50 g of alcohol or more in one single occasion and for women 40gr or more of alcohol.
[f]Sleep information was available for 37,536 participants.
Source data are provided as a Source Data file.

impact these results. Finally, the exclusion of prevalent cases of obesity, type 2 diabetes and apnea did not change the observed associations (Table S2).

Finally, we noticed that the time interval between the last meal and bedtime was inversely associated with the risk of overall CVD (Figure S6).

## Discussion

In this large prospective cohort study, later times of first and last meals were independently associated with a higher risk of overall CVD. These associations were stronger in women than in men. We observed no link between the daily number of eating occasions and the risk of overall CVD.

To the best of our knowledge, no prior prospective study has investigated the associations between specific meal timing, fasting and incident CVDs (hard endpoints). Mounting evidence suggests that the habit of regularly skipping breakfast[9–11,14,27–30] and eating late-at-night could have a negative impact for cardiometabolic health[12–14]. However, a recent meta-analysis found no impact of breakfast skipping on body weight[31]. The assessment of breakfast skipping and late-night eating is prone to classification bias, as described in the introduction. In line with our findings, a usual delayed first meal of the day (per 30-min and per quartile delay) has been associated in two observational studies with cardiometabolic risk factors (intermediate endpoints) including a worse cardiovascular health, measured with the American Heart Association score, and increased blood pressure, C-reactive protein concentration, insulin and glucose levels and lower high-density lipoprotein[15,16]. Similarly, a later time of last meal (per 1-h delay) was associated with higher HbA1c[16] all of which are risk factors for CVD.

### Time of day of food intake

Dietary behaviours are part of the main modifiable risk factors contributing to the global burden of CVD[32]. It is becoming more evident that the optimal metabolism of food is time-of-day-dependent[33]. Food is a well-known synchroniser of peripheral clocks in the circadian system which controls blood pressure rhythms[4] and eating late-at-night can disrupt this system and lead to metabolic disturbances[34]. Sensitivity to insulin and to elevated glucose concentration are greatest in the early morning and decline over the day, showing that metabolism is prepared to anticipate and digest energy sources at specific times of the day[35].

In animal models, delaying the first meal of the day by 4 hours increased body weight, hepatic lipids and adipose tissue weight and delayed circadian oscillation of genes related with lipid metabolism[36,37]. Mimicking late-night eating in mice has been also associated with phase alterations in peripheral clocks, weight gain, hepatic lipid accumulation, inflammation and microbial dysbiosis[38–40]. Evidence from RCTs suggests that a later evening meal can lead to glucose intolerance, insulin resistance, increased cholesterol and triglyceride levels and BMI[41,42]. Food intake when melatonin levels are high, during the rest phase, could lead to glucose intolerance and hyperglycaemia[43,44]. As seen from animal and human studies[36,37,42], having a later first and last meal of the day could be linked to CVDs through weight gain. However, in sensitivity analyses we adjusted our primary models for weight gain during follow-up and results did not substantially change, suggesting other mechanisms could be explaining these associations.

Other hypotheses could also explain our findings. First, night shift work has been associated with increased risk of cerebrovascular diseases; therefore, it is possible that the association is confounded by night shift and/or sleep disturbances[45,46]. However, in our study population, we excluded participants reporting extremely disrupted circadian nutritional behaviours, assuming these would correspond to individuals with severe circadian disruption, which is the case of night shift workers. We observed a minor attenuation after adjusting for chronotype, which might partly explain our results. Then, skipping

breakfast has been linked to a lower nutritional adequacy to dietary guidelines[8], but we controlled the present analyses for total daily energy intake, a healthy diet pattern and also for other nutritional indicators in sensitivity analyses such as consumption of ultra-processed foods. Moreover, it could be that individuals who are already obese and desire weight loss may be more likely to engage in breakfast skipping practices. However, we adjusted all our models for baseline BMI and also excluded participants with prevalent obesity in sensitivity analyses. The health status of the participants may also influence these associations but we explored adjusting for number of medications, as a proxy, and results were independent.

### Differences by sex
In our study, the association between late first and last meals and a higher overall CVD risk was stronger in women than in men. These differences could be linked to sexual dimorphisms in the anatomy and physiology of the circadian system[47–49]. Furthermore, this is similar to what has been reported in terms of circadian disruption due to night shift work and cardiovascular diseases[50]. In line with these results, late-night eating has been associated with arterial stiffness[12] and metabolic syndrome in women but not in men[13]. Additionally, a higher proportion of calories consumed in the evening has been also linked with increased levels of C-Reactive protein[51], and with increased BMI, waist circumference and blood pressure in women[52].

### The potential of time-restricted eating
We found an inverse association between nighttime fasting duration and the risk of cerebrovascular diseases. Our results on nighttime

**Table 2 | Association of meal timing and number of eating occasions with risk of cardiovascular diseases in the NutriNet-santé cohort, 2009-2021, _N_ = 103,389**

|  | N cases/ non-cases | HR (95% CI)[a] | p-val[b] |
|---|---|---|---|
| **Overall cardiovascular diseases** |  |  |  |
| Time of first meal (1 h incr.) | 2036/101,353 | 1.06 (1.01–1.12) | 0.02 |
| Before 8AM | 1040/45,266 | Ref. | 0.06 |
| Between 8 and 9AM | 764/36,217 | 1.07 (0.97–1.17) |  |
| After 9AM | 232/19,870 | 1.14 (0.98–1.32) |  |
| Time of last meal (1 h incr.) | 2036/101,353 | 1.02 (0.98–1.07) | 0.4 |
| Before 8PM | 786/33,937 | Ref. | 0.06 |
| Between 8 and 9PM | 844/44,766 | 1.08 (0.97–1.19) |  |
| After 9PM | 406/22,650 | 1.13 (0.99–1.29) |  |
| Number of eating occasions (1 occasion incr.) | 2036/101,353 | 0.99 (0.96–1.02) | 0.5 |
| **Cerebrovascular diseases[c]** |  |  |  |
| Time of first meal (1 h incr.) | 988/102,401 | 1.06 (0.98–1.14) | 0.1 |
| Before 8AM | 508/45,798 | Ref | 0.1 |
| Between 8 and 9AM | 361/36,620 | 1.02 (0.89–1.17) |  |
| After 9AM | 119/19,983 | 1.23 (0.99–1.52) |  |
| Time of last meal (1 h incr.) | 988/102,401 | 1.08 (1.01–1.15) | 0.02 |
| Before 8PM | 363/34,360 | Ref. | <0.01 |
| Between 8 and 9PM | 426/45,184 | 1.19 (1.03–1.37) |  |
| After 9PM | 199/22,857 | 1.28 (1.05–1.55) |  |
| Number of eating occasions (1 occasion incr.) | 988/102,401 | 0.97 (0.93–1.01) | 0.1 |
| **Coronary heart diseases[d]** |  |  |  |
| Time of first meal (1 h incr.) | 1071/102,318 | 1.05 (0.98–1.13) | 0.1 |
| Before 8AM | 550/45,756 | Ref. | 0.4 |
| Between 8 and 9AM | 406/36,575 | 1.09 (0.96–1.24) |  |
| After 9AM | 115/19,987 | 1.04 (0.84–1.29) |  |
| Time of last meal (1 h incr.) | 1071/102,318 | 0.97 (0.92–1.05) | 0.4 |
| Before 8PM | 432/34,291 | Ref. | 0.9 |
| Between 8 and 9PM | 429/45,181 | 0.99 (0.86–1.13) |  |
| After 9PM | 210/22,846 | 1.00 (0.83–1.20) |  |
| Number of eating occasions (1 occasion incr.) | 1071/102,318 | 1.01 (0.97–1.05) | 0.6 |

**Table 3 | Association between daily nighttime fasting duration and risk of cardiovascular diseases in the NutriNet-Santé cohort, 2009–2021, _N_ = 103,389**

|  | N cases/ non-cases | HR (95% CI)[a] | p-val[b] |
|---|---|---|---|
| **Overall cardiovascular diseases** |  |  |  |
| Continuous (1 h incr.) | 2036/101,353 | 0.98 (0.94–1.02) | 0.4 |
| 12 h or less | 1207/56,813 | Ref. | 0.6 |
| 12 h to 13 h | 558/28,380 | 0.91 (0.82–1.02) |  |
| More than 13 h | 271/16160 | 0.98 (0.84–1.14) |  |
| **Cerebrovascular diseases[c]** |  |  |  |
| Continuous (1 h incr.) | 988/102,401 | 0.93 (0.87–0.99) | 0.02 |
| 12 h or less | 613/57,407 | Ref. | 0.02 |
| 12 h to 13h | 254/28,684 | 0.78 (0.66–0.91) |  |
| More than 13 h | 121/16,310 | 0.80 (0.63–1.01) |  |
| **Coronary heart diseases[d]** |  |  |  |
| Continuous (1 h incr.) | 1071/102,318 | 1.03 (0.96–1.09) | 0.4 |
| 12 h or less | 613/57,407 | Ref. | 0.2 |
| 12 h to 13 h | 307/28,631 | 1.05 (0.90–1.22) |  |
| More than 13 h | 151/16,280 | 1.16 (0.94–1.44) |  |

_HR_ hazard ratio, _N_ sample size, _CI_ confidence Interval.
[a]Multivariable Cox proportional hazard models adjusted for age (timescale), sex (women, men), educational level (less than high school degree, <2 years after high school degree, ≥2 years after high school degree), monthly income per unit of consumption (<900€, 900–1200€, 1200–1800€, 1800–2300€, 2300–3700€, more than 3700€, do not want to answer), BMI at baseline (continuous, kg/m²), family history of CVDs (no, yes), alcohol consumption (Non-consumers (0 g/day), low consumers (0.1–4.9 g/day), moderate consumers (5.0–14.9 g/day), high consumers (15.0–29.9 g/day) and very high consumers (>30.0 g/day)), episodes of binge drinking (None, one, more than one), daily energy intake excluding alcohol (continuous, kcal/day), healthy and Western dietary patterns derived by factorial analysis (continuous), smoking (current regular (1 cigarette or more per day), current occasional, former, never), number of pack years (continuous, defined as the number of packs of cigarettes smoked per day by the number of years of smoking), physical activity (low, moderate, high) and number of dietary records (continuous). Time of first and last meal and number of eating occasions were mutually adjusted.
[b]P-values for continuous variables and _p_-value for trend for categorical variables.
[c]Stroke and transient ischemic attack.
[d]Myocardial infarction, acute coronary syndrome, angioplasty and angina pectoris.
Source data are provided as a Source Data file.

_HR_ hazard ratio, _N_ sample size, _CI_ confidence Interval.
[a]Multivariable Cox proportional hazard models adjusted for age (timescale), sex (women, men), educational level (less than high school degree, <2 years after high school degree, ≥2 years after high school degree), monthly income per unit of consumption (<900€, 900–1200€, 1200–1800€, 1800–2300€, 2300–3700€, more than 3700€, do not want to answer), BMI at baseline (continuous, kg/m2), family history of CVDs (no, yes), alcohol consumption (Non-consumers (0 g/day), low consumers (0.1–4.9 g/day), moderate consumers (5.0–14.9 g/day), high consumers (15.0–29.9 g/day) and very high consumers (>30.0 g/day)), episodes of binge drinking (None, one, more than one), daily energy intake excluding alcohol (continuous, kcal/day), healthy and Western dietary patterns derived by factorial analysis (continuous), smoking (current regular (1 cigarette or more per day), current occasional, former, never), number of pack years (continuous, defined as the number of packs of cigarettes smoked per day by the number of years of smoking), physical activity (low, moderate, high), number of dietary records (continuous), number of eating occasions (continuous) and time of first meal.
[b]P-values for continuous variables and _p_-value for trend for categorical variables.
[c]Stroke and transient ischemic attack.
[d]Myocardial infarction, acute coronary syndrome, angioplasty and angina pectoris.
Source data are provided as a Source Data file.

**Table 4 | Association of meal timing, number of eating occasions and nighttime fasting duration with risk of cardiovascular disease by sex in the NutriNet-santé cohort, 2009-2021, *N* = 103,389**

| | Women | | | Men | | |
|---|---|---|---|---|---|---|
| | N cases/non-cases | HR (95% CI)[a] | p-val[b] | N cases/non-cases | HR (95% CI)[a] | p-val[b] |
| **Overall cardiovascular diseases** | | | | | | |
| Time of first meal (1 h incr.) | 1106/80,603 | 1.06 (0.99–1.14) | 0.08 | 930/20,750 | 1.06 (0.99–1.15) | 0.09 |
| Before 8AM | 538/35,161 | Ref | 0.05 | 502/10,105 | Ref. | 0.6 |
| Between 8 and 9AM | 414/29,403 | 1.04 (0.91–1.19) | | 350/6814 | 1.09 (0.95–1.26) | |
| After 9AM | 154/16,039 | 1.24 (1.02–1.51) | | 78/3831 | 0.98 (0.76–1.25) | |
| Time of last meal (1 h incr.) | 1106/80,603 | 1.07 (1.00–1.13) | 0.05 | 930/20,750 | 0.97 (0.91–1.04) | 0.4 |
| Before 8PM | 387/16,039 | Ref. | 0.01 | 399/7040 | Ref. | 1.0 |
| Between 8 and 9PM | 479/36,347 | 1.12 (0.97–1.29) | | 365/8419 | 1.04 (0.89–1.20) | |
| After 9PM | 240/17,359 | 1.26 (1.05–1.51) | | 166/5291 | 0.99 (0.80–1.21) | |
| Number of eating occasions (1 occasion incr.) | 1106/80,603 | 0.99 (0.95–1.03) | 0.6 | 930/20,750 | 0.99 (0.95–1.03) | 0.7 |
| Nighttime fasting duration[c] (1 h incr.) | 1106/80,603 | 0.94 (0.88–1.00) | 0.05 | 930/20,750 | 1.03 (0.96–1.09) | 0.4 |
| **Cerebrovascular diseases[d]** | | | | | | |
| Time of first meal (1 h incr.) | 625/81,084 | 1.10 (1.01–1.21) | 0.03 | 363/21,680 | 0.98 (0.87–1.11) | 0.8 |
| Before 8AM | 299/35,400 | Ref | 0.02 | 209/10,398 | Ref. | 0.8 |
| Between 8 and 9AM | 242/29,575 | 1.11 (0.94–1.33) | | 119/7045 | 0.86 (0.68–1.08) | |
| After 9AM | 84/16,109 | 1.35 (1.04–1.75) | | 35/3874 | 1.00 (0.69–1.46) | |
| Time of last meal (1 h incr.) | 625/81,084 | 1.07 (0.98–1.16) | 0.1 | 363/21,680 | 1.09 (0.99–1.21) | 0.09 |
| Before 8PM | 218/27,066 | Ref. | 0.02 | 145/7294 | Ref. | 1.0 |
| Between 8 and 9PM | 275/36,551 | 1.18 (0.98–1.42) | | 151/8633 | 1.20 (0.94–1.52) | |
| After 9PM | 132/17,467 | 1.31 (1.03–1.67) | | 67/5390 | 1.20 (0.87–1.67) | |
| Number of eating occasions (1 occasion incr.) | 625/81,084 | 0.98 (0.93–1.04) | 0.6 | 363/21,680 | 0.94 (0.88–1.01) | 0.09 |
| Nighttime fasting duration (1 h incr.) | 625/81,084 | 0.94 (0.86–1.02) | 0.1 | 363/21,680 | 0.91 (0.82–1.01) | 0.09 |
| **Coronary heart diseases[e]** | | | | | | |
| Time of first meal (1 h incr.) | 495/81,214 | 0.99 (0.90–1.10) | 0.9 | 576/21,104 | 1.11 (1.01–1.22) | 0.03 |
| Before 8AM | 250/35,449 | Ref. | 1.0 | 300/10,307 | Ref. | 0.2 |
| Between 8 and 9AM | 175/29,642 | 0.93 (0.76–1.13) | | 231/6933 | 1.23 (1.04–1.47) | |
| After 9AM | 70/16,123 | 1.06 (0.79–1.42) | | 45/3864 | 0.97 (0.70–1.35) | |
| Time of last meal (1 h incr.) | 495/81,214 | 1.07 (0.97–1.18) | 0.2 | 576/21,104 | 0.91 (0.84–0.99) | 0.02 |
| Before 8PM | 174/27,110 | Ref. | 0.2 | 258/7181 | Ref. | 0.2 |
| Between 8 and 9PM | 212/36,614 | 1.06 (0.86–1.30) | | 217/8567 | 0.94 (0.78–1.14) | |
| After 9PM | 109/17,490 | 1.18 (0.90–1.55) | | 101/5356 | 0.87 (0.67–1.12) | |
| Number of eating occasions (1 occasion incr.) | 495/81,214 | 0.99 (0.93–1.05) | 0.6 | 576/21,104 | 1.03 (0.98–1.08) | 0.3 |
| Nighttime fasting duration (1 h incr.) | 495/81,214 | 0.93 (0.85–1.03) | 0.2 | 576/21,104 | 1.10 (1.01–1.19) | 0.02 |

*HR* hazard ratio, *N* sample size, *CI* confidence Interval.

Interactions between time of first meal and sex for the association with overall CVD, cerebrovascular diseases and coronary heart diseases were *p*-value = 0.2, 0.1 and 0.8 respectively. Interaction between time of last meal and sex for the association with overall CVD, cerebrovascular diseases and coronary heart diseases were *p*-value = 0.01, 0.6 and 0.004, respectively. Interaction between number of eating occasions and sex for the association with overall CVD, cerebrovascular diseases and coronary heart diseases were *p*-value = 0.3, 0.2 and 0.8, respectively. Interaction between nighttime fasting duration and sex for the association with overall CVD, cerebrovascular diseases and coronary heart diseases were *p*-value = 0.2, 0.5 and 0.02, respectively.

[a]Multivariable Cox proportional hazard models adjusted for age (timescale), educational level (less than high school degree, <2 years after high school degree, ≥2 years after high school degree), monthly income per unit of consumption (<900€, 900–1200€, 1200–1800€, 1800–2300€, 2300–3700€, more than 3700€, do not want to answer), BMI at baseline (continuous, kg/m$^2$), family history of CVDs (no, yes), alcohol consumption (Non-consumers (0 g/day), low consumers (0.1–4.9 g/day), moderate consumers (5.0–14.9 g/day), high consumers (15.0–29.9 g/day) and very high consumers (>30.0 g/day)), episodes of binge drinking (None, one, more than one) alcohol intake (continuous, kcal/day), daily energy intake excluding alcohol (continuous, kcal/day), healthy and Western dietary patterns derived by factorial analysis (continuous), smoking (current regular (1 cigarette or more per day), current occasional, current, former, never), number of pack years (continuous, defined as the number of packs of cigarettes smoked per day by the number of years of smoking), physical activity (low, moderate, high) and number of dietary records (continuous). Circadian nutritional behaviours were mutually adjusted (except for nighttime fasting duration).

[b]*P*-values for continuous variables and p-value for trend for categorical variables.

[c]The models for nighttime fasting duration were adjusted for the same variables as indicated in point 1 but excluding time of last meal of the day to avoid overadjustment.

[d]Stroke and transient ischemic attack.

[e]Myocardial infarction acute coronary syndrome, angioplasty and angina pectoris.

Source data are provided as a Source Data file.

fasting duration are in line with the growing body of evidence on TRE and its positive impact on markers of cardiometabolic health in humans such as reduced blood pressure, increased insulin sensitivity and reduced body weight[19–21]. There are also studies in animal models that suggest that restricting feeding to the active phase could prevent obesity and metabolic disturbances[17,18]. However, there are also discordant results; two observational studies reported an association between a longer nighttime fasting duration and higher prevalence of cardiometabolic risk factors[15,16]. These models investigating fasting duration were not adjusted for the time of the first meal, something that might be confounding these results. On the other hand, two short-term clinical trials (12-52 weeks) have shown no benefits for TRE on

weight loss in participants with overweight and obesity[25,53]. Similarly, a randomised crossover trial in 30 subjects with overweight or obesity, showed that a morning-loaded energy intake diet had no benefit on weight loss compared with an evening loaded diet[54].

Noticeably, the time of the day when the fasting period occurs could also play a role in cardiometabolic health. Considering evidence on the negative impacts of delaying the first and the last meal of the day, it is reasonable to think that it would be better to practice TRE by having an early first and last meal of the day, i.e., early-TRE (eTRE). Recent interventional studies are starting to demonstrate that eTRE is more beneficial than other fasting regimens with later eating windows throughout the day[55]. Moreover, this could explain the discrepant results in some studies on TRE with later eating schemes (eating window from 12PM to 8PM[25] or from 10AM to 7PM[24]). Interestingly, a study among participants of the NHANES survey showed that in women, extending the nighttime fasting duration was associated with lower levels of C-reactive protein level (but not with insulin resistance models) only among those who ate fewer than 30% of their calories after 5PM[51]. Although in our study population we did not find a statistically significant interaction between nighttime fasting and the time of the first meal of the day, our results on nighttime fasting duration, taken together with those on the time of first and least meals, are suggestive of a potential protective association between a longer nighttime fasting duration (i.e., TRE) and cardiovascular health, only if coupled with early first and last meals (i.e., eTRE), rather than by skipping breakfast. These findings are somehow in line with our previous results from the NutriNet-Santé cohort suggesting that nighttime fasting duration is only associated with a lower type 2 diabetes incidence in participants having breakfast before 8AM and fasting for >13 h overnight[56]. In another investigation conducted among participants of the Spanish multicase-control study[57], we also found that the association between nighttime fasting duration and prostate cancer risk was slightly modified by the time of the first meal. Among participants who had breakfast at 8:30 AM or earlier, a longer fasting period was associated with a lower risk of prostate cancer.

Finally, as regards meal frequency, and in line with our findings, no association was observed with the risk of coronary heart disease in the Males Health Professionals Follow-up Study (HPFS)[14].

## Strengths and limitations

The sample size, prospective design and detailed assessment of circadian dietary behaviours encompass the main strengths of this study. These behaviours were measured using dietary records, which are less subject to recall and misclassification bias than are dietary recalls or ad-hoc questionnaires. The large panel of questionnaires in the NutriNet-Santé cohort enabled us to control for a large number of well-measured potential confounders, reducing the risk of confounding in the present analyses. Besides, the observational, yet prospective design has allowed us to study the long-term associations (follow-up from 2009 to 2021) between meal timing and cardiovascular diseases, which would be challenging to investigate in interventional studies, especially with hard endpoints.

However, several limitations have to be acknowledged. Sleep time and duration were assessed among a sub-sample during follow-up, and therefore could not be used in the main analyses: this sub analysis might have been subject to selection bias and loss of statistical power. Similarly, chronotype was assessed in this same questionnaire and it was self-reported by the participants, however, specific questionnaires such as the morningness-eveningness questionnaire (MEQ) could have been more appropriate tools. Furthermore, night shift work was approximated from the sleep times reported in this follow-up questionnaire, since information on current and history of shift work were not available in the cohort. Although we accounted for a large panel of confounders, given the observational nature of this study, residual confounding cannot be completely ruled out. We had no information

on exposure to light-at-night, use of recreational drugs, as well as timing of physical activity, medication or alcohol consumption which are all potential disruptors of circadian rhythms. Future studies collecting accelerometry data could be of interest to objectively account for physical activity timing and patterns. Furthermore, potential measurement imperfections in collected data despite adjustment (e.g., job-related factors, chronotype) as well as other unknown or unmeasured potential confounders (e.g., deliberate choice of meal timing and awakening, partying) could be contributing to residual confounding in this study. For instance, data about natural vs. alarm clock awakening or children waking their parents up were not available in our cohort, and might have introduced unmeasured confounding, since the method of awakening could influence sleep inertia and quality, and could therefore have an impact on hunger and meal timing[58,59]. We used a falsification endpoint that is not expected to be related to the exposure in a way to reduce the risk of spurious findings in our main models. In these analyses, our exposures of interest were not associated with the negative control (basal cell carcinoma). Moreover, reverse causation bias linked to change of behaviours in participants with poor health having difficulty getting out of bed in the mornings because of health problems, cannot be entirely ruled out, despite efforts made in the sensitivity analyses.

Participants in the NutriNet-Santé cohort are volunteers and are more likely to be women, have a higher socioeconomic status and healthier behaviour patterns than the general population, somehow limiting the extrapolation of these results[60]. Moreover, the healthier behaviours in the study population could have led to a lower incidence of CVD compared to the general population and to an underestimation of the studied associations, even though an overestimation cannot be totally excluded, as women, overrepresented in our study, tended to have later times of first meal than men. Next, nutritional behaviours were averaged from all available dietary records, during the first two years of follow-up, which included data on working and non-working days. This omitted the variability in circadian nutritional behaviours between working and non-working days, a concept that has been previously named as eating jet lag[61]. However, in sensitivity analyses we adjusted for eating jet lag and our associations of interest were not changed. Further studies are required to explore whether this variability in timing could also be associated with CVD.

Lastly, even though meal timing showed associations with cardiovascular outcomes in our study, CVD remains a multifactorial disease, and meal timing alone could not explain the trends in CVD incidence across countries, as other demographic, lifestyle, genetic and environmental factors are involved.

To conclude, in this large prospective study, later times of first and last meals were associated with a higher risk of overall CVD. This work, which needs replication in other large-scale cohorts in different settings and using different and complementary approaches to minimise residual confounding and selection bias, supports an important role of adopting earlier eating timing patterns, consistently with previous experimental and observational studies. These findings suggest that, beyond the nutritional quality of the diet itself, recommendations related to meal timing for patients and citizens may help promoting a better cardiometabolic health.

## Methods

### The NutriNet-Santé cohort

The NutriNet-Santé cohort study was launched in France in 2009 and aimed to better understand the relationship between nutrition and health. This is an ongoing web-based cohort study that targets volunteers aged 18 or older recruited through various multimedia channels (https://etude-nutrinet-sante.fr/). The protocol of the study has been published previously[62]. All participants are required to provide an electronic informed consent when enrolling the study. The NutriNet-Santé study is conducted according to the Declaration of

Helsinki guidelines and is registered at clinicaltrials.gov (NCT03335644). The study was approved by the Institutional Review Board of the French Institute for Health and Medical Research (IRB Inserm no 0000388FWA00005831) and the "Commission Nationale de l'Informatique et des Libertés" (CNIL n°908450/n°909216). Confidentiality and security of the data are assured at all time.

At inclusion and yearly, participants are invited to complete a set of questionnaires including data on socio-demographics and lifestyle[63], physical activity (through a validated 7-day assessment, the International Physical Activity questionnaire - short form [IPAQ][64]), anthropometrics[65,66], health status and diet (details below). In 2014, a subset of the study population responded to an optional comprehensive sleep questionnaire, which included a question on chronotype (Do you consider yourself as: clearly a morning person, rather a morning person, clearly an evening person, rather an evening person, neither morning or evening, both or I don't know)[67].

## Dietary assessment

In the NutriNet-Santé cohort study, diet was assessed from baseline with bi-annual series of 3 random records of 24h food intake (including information on a non-working day). Participants had to report all foods and beverages consumed and also when during the day they consumed them. In case participants could not provide the specific quantity, they were asked to estimate portion sizes through validated photographs or usual containers[68]. These questionnaires have been validated against biomarkers[69,70] and an interview by a dietician[71]. Baseline dietary intakes were assessed by averaging the consumption over the food records available from the two first years of follow-up, considered as the habitual dietary behaviours, and merged with French food composition database with more than 3,500 items[72]. We estimated mean daily intakes of energy, alcohol, macro- and micronutrients for this period. We used basal metabolic rate and the Goldberg cut-off method[73] to identify and exclude energy under-reporters. Details of this procedure are explained in the Supplementary Material, Method A.

To estimate the time of first and last meal of the day and the number of eating occasions, we computed an average of the food records available from the first two years of follow-up and we included only participants with at least 3 dietary records. An eating occasion was defined as the intake of any food or beverage of at least 1 kcal, thus excluding water drinking but including all other eating episodes. We have made this assumption as we wanted to be conservative by capturing any eating occasion without a predefined definition (including, for example, a coffee with milk or other caloric or low-caloric drinks). We also decided to exclude participants reporting the consumption of a first meal after 3PM or a last meal before 3PM since they corresponded to very disrupted circadian behaviours (e.g. night shift workers). Finally, we calculated nighttime fasting duration as follows: 24h minus the time elapsed between the first and the last meal of the day.

## Ascertainment of cardiovascular disease events

Major health events, including CVD, were self-reported by the participants through a health questionnaire sent every 6 months, or through a permanently available platform dedicated to health events collection on the NutriNet-Santé website. Additionally, participants were asked to confirm the declaration of any health event with medical records including diagnoses, examinations or hospitalisations. Physicians from the research team reviewed the medical records and validated health events and in case of any doubt contacted the physician of the participant. Similarly, participants' families or doctors were contacted if there was no response in the website for more than a year. CVD events were complemented with the linkage to medico-administrative databases of the national health insurance (SNIIRAM) that include information on prescription medication and medical consultation and hospitalisation history. Finally, deaths were assessed with the linkage to the French national cause-specific mortality registry (CépiDC). We used the International Classification of Diseases-Clinical Modification codes (ICD-CM, 10th revision) to classify CVD events. In the present study, we considered incident cases of stroke (I64), transient ischemic attack (G45.8, G45.9), myocardial infraction (I21), angina pectoris (I20.1, I20.8, I20.9), acute coronary syndrome (I20.0, I12.4) and angioplasty (Z95.8). Sixty percent of the angina cases had at least one cardiovascular risk factor (hypercholesterolemia, overweight/obesity, smoking, diabetes, and/or hypertension). We classified stroke and transient ischemic attack as cerebrovascular diseases and myocardial infraction, angina pectoris, acute coronary syndrome and angioplasty as coronary heart diseases.

## Statistical analyses

As of October 5th 2021st, 103,389 participants without prevalent CVD at baseline were included in the present analyses. A detailed flowchart is presented in Figure S1. Time of first and last meal of the day and number of eating occasions were considered as continuous (per hour increase and per meal increase, respectively) and categorical variables, as an approximation of the tertiles in the general population. We examined the correlation among these nutritional behaviours with Spearman rank (Figure S2). We built cause-specific Cox proportional hazards models to estimate hazard ratios (HR) and 95% confidence intervals (CI) for the associations of meal timing and number of eating occasions with risk of developing overall CVD, cerebrovascular diseases and coronary heart disease. Since we aimed to investigate the risk of the first cardiovascular event, we used a competing risks approach by applying cause-specific Cox models: for instance, in models investigating cerebrovascular diseases, if a coronary heart disease was diagnosed during the follow-up, the participant was censored at their date of coronary heart disease diagnosis and was considered as non-case for cerebrovascular diseases, and the same rationale was applied the other way around. Therefore, participants contributed to person-time until the date of CVD diagnosis, censoring event, last connection or October 5th 2021 whichever occurred first. We confirmed the Cox models' assumption of risk proportionality using Schoenfeld residuals, and the assumption of linearity by introducing spline terms for each of the exposure variables in the models (Figs. S4 and S5).

We adjusted our principal models for age (timescale), sex (women, men), education (no high-school degree, less than 2 years of education after high-school, 2 years or more of high school education), monthly income per unit of consumption (<900€, 900–1200€, 1200–1800€, 1800–2300€, 2300–3700€, more than 3700€, do not want to answer), BMI at baseline (continuous, kg/m$^2$), family history of CVD (yes, no), alcohol consumption (Non-consumers [0 g/day], low consumers [0.1–4.9 g/day], moderate consumers [5.0–14.9 g/day], high consumers [15.0 −29.9 g/day] and very high consumers [>30.0 g/day][74]), episodes of binge drinking defined as 50gr of alcohol or more in one single occasion for men and 40gr for women[75] (None, one, more than one in the baseline records), energy intake excluding alcohol (continuous, kcal/day), smoking (current regular [1 cigarette or more per day], current occasional [<1 cigarette per day], former, never), number of pack years (continuous, defined as the number of packs of cigarettes smoked per day multiplied by the number of years of smoking), physical activity level calculated according to IPAQ recommendations (low, intermediate, high)[64], number of dietary records (continuous), healthy dietary pattern and Western dietary pattern. The two pattern variables were calculated by principal component analysis (PCA) on the basis of 20 predefined food groups (further details are provided in Supplementary Material, Method B). Finally, the time of first daily meal, time of the last daily meal and number of eating occasions were mutually adjusted. We checked the variance inflation factor to control for redundancy. All covariates had less than 1% of

missing data except for physical activity which had 14%. We performed multiple imputation on missing values by chained equations (MICE)[76] and combined the results from 20 complete datasets using the method proposed by Rubin[77].

We then explored the hypothesis of an extended period of fasting as a protective factor for CVD. Nighttime fasting duration was investigated as a continuous variable in hours and as a categorical variable (1) 12 hrs or less, 2) 12 to 13 h or 3) more than 13 h). This was categorised on the basis of the eating/fasting schemes most frequently reported in the literature in relation with time-restricted eating[19]. Models were adjusted for the same covariates as indicated above, including time of first meal, but not time of last meal to avoid collinearity.

We investigated the modulating role of time of first meal in the association between nighttime fasting duration and risk of CVD. This was performed by introducing an interaction term between nighttime fasting duration and time of first meal of the day. The significance of the interaction term was tested using a likelihood ratio test.

Lastly, we explored interactions with sex, BMI at baseline, menopausal status among women ($N = 81,709$) and chronotype, i.e., the individual preference for timing of activities during the day (morning, intermediate, evening)[78], among those who responded to the sleep questionnaire (see details below). Indeed, we wanted to explore if the associations between eating patterns and CVD risk were different according to different BMI categories, as suggested previously[79]. Despite significant interactions with sex, overall results were presented first to provide an overview of the general trends, before presenting sex-specific results, for readability purposes. As regards chronotype, there were also previous investigations showing that this individual preference could modify the association between circadian alterations and health in humans[80,81]. Finally, we explored the interaction with menopausal status among women as some differences have also been reported in relation to breast cancer risk[82].

### Sensitivity analyses

In sensitivity analyses, we explored further adjustment of our models for well-known individual nutrients related with CVD risk including daily consumption of saturated fatty acids, sodium, sugar, red and processed meat, sugary drinks, fruits and vegetables, nuts, whole grain, yoghurt and ultra-processed foods[32]. We also considered the French region of residence of the participants (to account for latitude), and profession. We also adjusted for weight change during follow-up, which was calculated as the percentage of weight change from baseline to end of study and divided by number of years of follow-up, to investigate whether the association was explained by weight change. We also adjusted for eating jet lag, reflecting the discrepancy in meal timings between working and non-working days, as this phenotype has been suggested to be associated with negative cardiometabolic outcomes[52]. This was defined, according to previously described methodology[61], as the difference between the eating midpoints (middle time point between first and last eating episode) of non-working and working days.

To challenge our models for reverse causation bias, we excluded cases diagnosed during the first two years of follow-up. We performed further adjustments for potential confounders such as prevalent cases of type 1 and 2 diabetes, hypercholesterolemia, hyperglyceridaemia and hypertension at baseline. For a subset of the study population, data from the sleep questionnaire was available. We explored restricting our analyses to this subpopulation and then we excluded participants reporting a bed time between 8AM and 6PM (to exclude potential night shift workers, N = 37,536). We then adjusted for sleep duration (h /24 h) and chronotype[78]. Furthermore, we accounted for the time interval between the last meal of the day and bedtime, as a shorter period could be linked with sleep disturbances[83] and, therefore, potentially negative cardiovascular outcomes[84,85]. We also

explored adjusting for number of awakenings during the night and for sleep apnea as an indicator of sleep quality.

We built a model without excluding participants with very extreme meal timings (having a first eating occasion after 3PM or a last eating occasion before 3PM). Additionally, we explored adjusting our models for marital status and number of children, as these factors can impact the sleep-wake cycle, as well as for number of medications, as participants might have to adapt their meal timing according to medication timing. Adjustments for the season of the physical activity questionnaire, as well as the season of the first 24-hour dietary record were also performed as physical activity practices might fluctuate throughout the year, and meal timing might exhibit seasonal changes. In the dietary records, participants were asked whether their 24-hour record was representative of a regular day or whether it was impacted by any extraordinary circumstances, such as illness or unusual events. We adjusted for this information. We also adjusted for the practice of caloric restriction or restrictive diet. Moreover, we performed models where we excluded from the sample participants with prevalent obesity, type 2 diabetes, or sleep apnea. Finally, as a falsification endpoint and to rule out spurious findings linked to chance, we examined the associations of our main exposures with basal cell carcinoma incidence.

All tests were two-sided, and we considered $P < 0.05$ to be statistically significant. R version 4.1.3 (The R Foundation for Statistical Computing Platform) was used for these analyses. Data and code will be available upon request.

### Reporting summary

Further information on research design is available in the Nature Portfolio Reporting Summary linked to this article.

## Data availability

Source data allowing reproducing figures and tables are provided with this paper. Raw data described in the manuscript are protected and are not available due to data privacy laws according to French regulations. Data can be made available upon request pending application and approval. Researchers from public institutions can submit a collaboration request including information on the institution and a brief description of the project to collaboration@etude-nutrinet-sante.fr. All requests will be reviewed by the steering committee of the NutriNet-Santé study within 8 to 12 weeks. If the collaboration request is accepted, a data access agreement will be necessary and appropriate authorisations from the competent administrative authorities may be needed. In accordance with existing regulations, no personal identification data will be accessible. The NutriNet-Santé food composition database is available in the book "Table de composition des aliments, Etude NutriNet-Santé, Editions Inserm – Economia", ISBN-10: 2717865373 ISBN-13: 978-2717865370.

## Code availability

Code book, and analytic code will be made available upon request pending application and approval, by sending an email to collaboration@etude-nutrinet-sante.fr.

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

## Acknowledgements

We thank, Cédric Agaësse, Alexandre De-Sa and Rebecca Lutchia (dietitians), Younes Esseddik (IT manager), Nathalie Druesne-Pecollo, PhD (operational coordinator), Thi Hong Van Duong, Régis Gatibelza, Jagatjit Mohinder and Aladi Timera (computer scientists); Fabien Szabo de Edelenyi, PhD (manager), Julien Allegre, Nathalie Arnault, Laurent Bourhis, and Nicolas Dechamp (data-manager/statisticians); Merveille Kouam and Paola Yvroud (health event validators) and Maria Gomes (participant support) for their technical contribution to the NutriNet-Santé study. We warmly thank all the volunteers of the NutriNet-Santé cohort. The NutriNet-Santé study is supported by the following public institutions: Ministère de la Santé, Santé Publique France, Institut

National de la Santé et de la Recherche Médicale (INSERM), Institut national de recherche pour l'agriculture, l'alimentation et l'environnement (INRAE), Conservatoire National des Arts et Métiers (CNAM) and Université Sorbonne Paris Nord. A.P.-C., D.R., and M.K. received support from the Spanish Ministry of Science and Innovation and State Research Agency through the "Centro de Excelencia Severo Ochoa 2019-2023" Program (CEX2018-000806-S), and support from the Generalitat de Catalunya through the CERCA Program. A.P.-C. is supported by a MINECO (Ministry of Economy in Spain) fellowship (PRE2019-089038). Researchers were independent from funders. Funders had no role in the study design, the collection, analysis and interpretation of data, the writing of the report, and the decision to submit the article for publication.

## Author contributions

The authors' contributions were as follows – A.P.-C., B.S. and M.T.: designed the research; V.A.A., L.F., C.J., A.B., E.K.G., S.H. and M.T.: collected the data; A.P.-C.: performed statistical analyses; B.S.: supervised statistical analyses; A.P.-C.: drafted the manuscript; B.S. and M.T.: supervised the writing; D.R. and M.K. participated to the supervision of the writing; M.T. and B.S. equally contributed and are co-last authors; all authors contributed to data interpretation, revised each draft for important intellectual content, read and approved the final manuscript. A.P.-C., B.S., and M.T. had full access to all the data in the study, B.S. takes responsibility for the integrity of the data and the accuracy of the data analysis, he is the guarantor. The corresponding author (B.S.) attests that all listed authors meet authorship criteria and that no others meeting the criteria have been omitted.

## Competing interests

The authors declare no competing interests.

## Ethical approval

NutriNet-Santé is conducted according to the Declaration of Helsinki guidelines and was approved by the institutional review board of the French Institute for Health and Medical Research (IRB Inserm n 0000388FWA00005831) and the Commission Nationale de l'Informatique et des Libertés (CNIL n 908450/n 909216). Electronic informed consent was obtained from all included participants in this study and could be withdrawn at any point of the study.
