## [Peer Review File · Nature Communications]

GENERAL COMMENT

Apart from what we eat, factors related to eating habits such as meal patterns, fasting periods and number of eating episodes may be important in relation to CVD risk. Little is known about these factors. The present manuscript is important from both a public health and clinical perspective. A large, unique dataset was used that allowed for detailed analyses, including sensitivity analyses. The paper is well-written.

However, I do have major concerns with this paper, including the following issues that may impact causal inference:

- residual confounding by socioeconomic status (e.g. job-related factors) and/or unemployment or lack of daytime activities
- residual confounding by marital status and household factors (are meal times chosen deliberately, or do people have to wake up at certain times because of children, jobs, studies?)
- the role of intermittent fasting, slimming diets, and other prescribed diets, which is not accounted for in the analyses
- the role of medication (type, dose, timing, with meals or in-between)
- reverse causation bias, e.g. people with poor health who have difficulty getting out of bed in the mornings because of health problems, poor sleep quality etc.
- residual confounding by behavioral factors that make people stay up till late, e.g. partying, alcohol intake pattern and the independent role these factors have on CVD risk
- residual confounding by season, which affect circadian rhythms, sleeping times, meal patterns, and also risk of CVD.

SPECIFIC COMMENTS

INTRODUCTION

line 57

“Breakfast skipping in observational studies is associated with more obesity”. The reason could be that people who want to lose weight skip breakfast (reverse causation bias). How is that accounted for in these studies?

METHODS

Line 89 and further

Was use of medication and recreational drugs recorded, including timing and frequency, and how were these data dealt with in the statistical analysis? Could drug use have an impact on the associations between meal timing and CVD?

Line 107

Why was only the level studied, rather than physical activity patterns (time of the day)? 7-d records on consecutive days? Were seasonal influences?

Line 111

3-d random food records. Were seasonal differences accounted for? How did the authors deal with sickness, people who are bedridden?

Line 127

An eating occasion was defined as the intake of any food or beverage of at least 1 kcal. What about the intake of medication (that also delivers some calories, and can be taken with a caloric drink)? What about intake of non-caloric artificially sweetened beverages and chewing gum?

Line 129-130

The authors excluded participants reporting the consumption of a first meal after 3PM or a last meal before 3PM since they corresponded to very disrupted circadian behaviours. Is that always true? People having their last meal at 3PM and first meal at 7AM may be intermittent fasters. Why not included as a confounder in the analysis, instead of excluding these people? How were people with intermittent fasting identified/analysed? Were people on a weight loss diet or other prescribed diet excluded?

Line 131-133

The authors calculated nighttime fasting duration as follows: 24h minus the time elapsed between the first and the last meal of the day. —> intakes of >1 kcal were considered eating episodes. So if people just had 1 candy (e.g. for sore throat) during the night, was that considered as an eating episode or disrupted fast?

Line 153-155

Did the authors consider sleep apnea as CVD risk factor, and if so, what was the impact of this factor on the associations?

Line 161-162

Please show the distributions for eating times and frequency. Data are not normally distributed. Is it justified to analyse these data continuously?

Line 169-170

How did the authors deal with unstable AP and TIA in their analyses of CHD and stroke, respectively?

Line 180-196

-Why adjusted for absolute alcohol intakes, rather than alcohol consumption patterns? The latter would be more relevant when studying meal timing. The same holds for physical activity.

-Were sleep duration and quality, and number of awakening episodes measured? Do the authors consider these factors as confounders or intermediary factors? Could they include these variables in their models for people who had these data?

-BMI is considered a confounder, and is part of all multivariable models. However, BMI could also be an intermediary factor, as well as an effect modifier. Would it not be better to stratify for BMI and/or to include it in sensitivity analysis?

-Was the time between awakening and first meal recorded? If so, should that variable not be included in the model?

-Were data collected on marital status? What is the impact of being married/single/widowed etc and having children on eating patterns and risk of CVD? Social factors (adaptation to household members) could play a major role in meal patterns, and are also related to CVD risk.

Line 245-260

Did the authors consider sensitivity analyses after exclusion of (cardiovascular, psychotropic or other) drug users?

Idem, after exclusion of people with disturbed sleeping patterns and/or sleep apnea?

Idem, after exclusion of people with (morbid) obesity or diabetes?

RESULTS

Table 1: People with lower education tend to have their breakfasts earlier in the morning. The opposite is true for income: those with lower incomes have their breakfasts later. I would not expect that. Have data been collected on type of job? What about unemployment or sickness? It is well-known that people with a lower socio-economic position (SEP) have a higher risk of CVD. Could residual confounding have impacted the conclusion of this study? Since SEP is one of the most important confounding factors in this study, the authors should further explore this (e.g. in sensitivity analyses).

Table 1: include information on marital status, medication use and sleep parameters. Consider these factors in the analysis.

Table 1: Why are men underrepresented in the analyses? How is that related to job activities, taking care of household, children etc.? Please explain and consider potential selection bias in the results.

Table 4: Because of effect modification, sex-specific analyses as presented in Table 4 should be considered as primary analyses rather than the total cohort. The number of men is less than the number of women, but still sufficiently large for sex-specific analyses.

DISCUSSION

-Discuss whether awakening occurred naturally or otherwise, by alarm clock or children waking up parents. If no data are available on these factors, discuss the potential impact based on the literature.

-Elaborate on the role of intermittent fasting and the results from this study. It is not yet clear to me whether abstaining from food (and/or beverages) after dinner until breakfast is recommended or not, and how long that fasting period should last. Periods of 12 hours or even 16 hours are recommended. For a country like the Netherlands, the latter would imply a fasting period from e.g. 7PM to 11AM. Do the authors consider that unhealthful, and why?

-Discuss residual confounding more thoroughly. See general comments.

Line 398-400

The authors state that "The association between late first and last meals and a higher overall CVD risk was stronger in women than in men. These differences could be mainly explained by the higher proportion of women in our cohort." This reasoning is not correct. Sample size has an impact on precision (confidence intervals), not on the measure of association (relative risks).

Line 408

If BMI could be part of the biological pathway for meal timing in relation to CVD, why did the authors adjust for it in their multivariable models?

Reviewers' Comments:

Reviewer #1:

Remarks to the Author:

Reviewer #2:

Remarks to the Author:

This is well written manuscript that examines how meal timing and nighttime fasting are related to cardiovascular health parameters. The topic is novel and will be of interest to the readers of this journal. The methods are sound, and the results are not overstated in the conclusion. I only have a few minor comments:

1. Chronotype was not measured in the prospective. It would have been important for chronotype to be recorded for each subject by the morningness-eveningness questionnaire (MEQ). Please comment on how not using the MEQ may have limited your findings.

2. Line 351 – Please also comment on the BMJ meta-analysis that reports no negative effect of breakfast skipping on body weight (<https://www.bmj.com/content/364/bmj.l42>) As it stands, this sentence is only reporting the negative outcomes of breakfast skipping, and lacks equipoise.

3. In the discussion, please comment on why some countries (e.g., Spain, Argentina, Chile, etc) that typically eat dinner late in the evening (10-11pm) do not have higher risk of CVD compared to countries (USA, Canada, France, etc) that typically eat dinner earlier in the evening (6-8 pm).

Reviewer #1 (Remarks to the Author):

General comment

Apart from what we eat, factors related to eating habits such as meal patterns, fasting periods and number of eating episodes may be important in relation to CVD risk. Little is known about these factors. The present manuscript is important from both a public health and clinical perspective. A large, unique dataset was used that allowed for detailed analyses, including sensitivity analyses. The paper is well-written.

However, I do have major concerns with this paper, including the following issues that may impact causal inference

- **Residual confounding by socioeconomic status (e.g. job-related factors) and/or unemployment or lack of daytime activities**
- **Residual confounding by marital status and household factors (are meal times chosen deliberately, or do people have to wake up at certain times because of children, jobs, studies?)**
- **The role of intermittent fasting, slimming diets, and other prescribed diets, which is not accounted for in the analyses**
- **The role of medication (type, dose, timing, with meals or in-between)**
- **Reverse causation bias, e.g. people with poor health who have difficulty getting out of bed in the mornings because of health problems, poor sleep quality etc.**
- **Residual confounding by behavioral factors that make people stay up till late, e.g. partying, alcohol intake pattern and the independent role these factors have on CVD risk**
- **Residual confounding by season, which affect circadian rhythms, sleeping times, meal patterns, and also risk of CVD.**

We thank Reviewer 1 for this careful revision. The points that have been raised are interesting and we have addressed all the suggestions. The major concerns commented above are discussed below in each specific comment. We agree with the reviewer that, as every observational study, residual confounding is a challenge to take into account in the interpretation of the results. Our approach of evaluating meal timing using an observational cohort study has the important advantage of evaluating the exposure (meal timing) in real life conditions that occur for a prolonged period of time. This is not the case in (most) human intervention or animal experiments that are usually short-term studies and that, obviously, have other advantages. We had already controlled for numerous factors in our analysis. The reviewer suggests several more factors that could possibly bias our results, and these are excellent suggestions. We provide new analyses adjusting for these factors and results remain largely unchanged. We provide detailed responses here below.

Specific comments

1. **Line 57. “Breakfast skipping in observational studies is associated with more obesity”. The reason could be that people who want to lose weight skip breakfast (reverse causation bias). How is that accounted for in these studies?**

We thank the reviewer for raising this point. We acknowledge that individuals who are already overweight and desire weight loss may be more likely to engage in breakfast skipping practices, particularly within the context of trends such as intermittent fasting. In the systematic review and

meta-analysis mentioned in the introduction and highlighted by the reviewer (line 57), 45 observational studies (36 cross-sectional studies and 9 cohort studies) were included. From the cohort studies, 4 were prospective. These prospective cohort studies recruited participants without obesity and followed them over time, collecting data on breakfast habits and other relevant variables. This design allows for the examination of the association between breakfast skipping and future obesity risk while minimizing reverse causation bias. In the sub meta-analysis of these 4 prospective cohort studies, the lowest frequency breakfast eating per week vs. the highest was significantly associated with a higher risk of overweight/obesity (Risk ratio [RR] = 1.35, 95% CI 1.22 – 1.49).

These studies can help establish the temporal sequence and provide stronger support for the hypothesis that breakfast skipping may contribute to obesity development rather than being a consequence of pre-existing obesity, even though reverse causality cannot be entirely ruled out, as some participants at baseline in these studies might have been metabolically unhealthy with normal weight, and therefore trying to change their behaviours

In our analysis, to address the potential influence of reverse causation bias, we controlled for baseline body mass index (BMI) as a potential confounding variable. By including baseline BMI as a covariate in our statistical models, we aimed to account for the initial weight status of participants and minimize the impact of pre-existing obesity on the association between breakfast skipping and obesity outcomes. In reviewer figure 1 we present the DAG for this association.

Reviewer figure 1. DAG for the association between obesity (or overweight), breakfast skipping and risk of developing CVDs during follow-up.

Moreover, in Reviewer table 5 in comments below, we additionally excluded prevalent cases of obesity at baseline and results remained unchanged. We have added this new analysis in the methods (p. 10, l. 279): “Moreover, we performed models where we excluded from the sample participants with prevalent obesity, type 2 diabetes, or sleep apnea”, the results (p. 13, l. 371): “Finally, the exclusion of prevalent cases of obesity, type 2 diabetes and apnea did not change the observed associations (Table S2)” and the discussion (p. 15, l. 430): “Moreover, it could be that individuals who are already obese and desire weight loss may be more likely to engage in breakfast skipping practices. However, we adjusted all our models for baseline BMI and also excluded participants with prevalent obesity in sensitivity analyses.”

2. Line 89 and further. Was use of medication and recreational drugs recorded, including timing and frequency, and how were these data dealt with in the statistical analysis? Could drug use have an impact on the associations between meal timing and CVD?

We thank the reviewer for raising this important question regarding the recording of medication and recreational drug use, and how these factors were handled in our statistical analysis. In our

study, the use of medication (name and dosing) at the moment of the enrolment was indeed collected in the baseline questionnaire (and repeated yearly during follow-up). However, medication timing was not collected, as it was not part of the 24-hour dietary records. As a result, we were unable to incorporate medication timing into our statistical analysis.

Nevertheless, we have now adjusted for the number of medications reported at baseline, as a proxy for overall medication usage. We present these new results in Reviewer table 1 and in Supplementary material (Table S1, Model 14). Overall, we observe that results remained mostly unchanged, indicating that these findings are independent of medication use. We have indicated this in the discussion (p. 15, l. 434): *“The health status of the participants may also influence these associations but we explored adjusting for number of medications, as a proxy, and results were independent.”*

Recreational drugs were not recorded in this study. However, it is worth noting that the consumption of recreational drugs often correlates with behaviours such as binge drinking, which may occur in specific social environments (e.g., party settings). In our main statistical models, we have adjusted for number of binge drinking episodes as a potential confounding factor. Models were also adjusted for amount of alcohol consumption, smoking and number of pack years, all of these usually being exacerbated as well in party settings. We have now added as a limitation in the discussion the fact that information about other recreational drugs such as cannabis, was not recorded. The text reads as follows (p. 17, l. 517): *“ We had no information on exposure to light-at-night, use of recreational drugs, as well as timing of physical activity, medication or alcohol consumption which are all potential disruptors of circadian rhythms.”*

3. Line 107. Why was only the level studied, rather than physical activity patterns (time of the day)? 7-d records on consecutive days? Were seasonal influences?

We thank the reviewer for highlighting these important considerations. In our study, we focused on assessing the overall level of physical activity rather than specific patterns or timing of physical activity throughout the day. The primary aim of our research was to investigate the association between meal timing and CVD risk, with physical activity level serving as one of several potential confounding factors. We recognize the importance of capturing comprehensive physical activity patterns and considering the variability of physical activity across different times of the day as a potential synchronizer of the circadian system. Unfortunately, data on timing of physical activity is not available for the moment in the NutriNet-Santé cohort. We were therefore not able to include it in our models.

We have mentioned this point as a limitation in the discussion (p. 17, l. 517): *“We had no information on exposure to light-at-night, use of recreational drugs, as well as timing of physical activity, medication or alcohol consumption which are all potential disruptors of circadian rhythms. Future studies collecting accelerometry data could be of interest to objectively account for physical activity timing and patterns. ”*

Additionally, considering seasonal influences on physical activity is valuable, as activity levels may fluctuate due to factors such as weather conditions, daylight duration, and seasonal activities. We have now adjusted for the season when the physical activity questionnaire was completed as this might have influenced their response. We present these new results in Reviewer table 2 and in supplementary material (Table S1, Model 15). Results were not modified, suggesting that this variable was not confounding our results. We have added this in the results (p. 13, l. 367):

“Similarly, adding the season of completion of the physical activity questionnaire and the first set of dietary records did not alter our results (Model 15 and 18)”

4. Line 111. 3-d random food records. Were seasonal differences accounted for? How did the authors deal with sickness, people who are bedridden?

In the NutriNet-Santé study, 24-hour food records were repeated every 6 months to account for potential seasonal variations in dietary intake. By collecting data throughout the year, the aim was to capture the seasonal diversity in dietary habits that may be influenced by factors such as the availability of certain foods and changes in eating patterns and daylight length. In line with this assumption, we have now added a sensitivity analysis where we adjust for the season of the first set of dietary record (supplementary material table s1, model 18 and methods p. 10, l. 272). The results remained unchanged.

Regarding participants who may be bedridden or experiencing illness, the NutriNet-Santé study implemented a system where participants could request alternative recording days up to three times if they were unable to complete the food records due to sickness or other unforeseen circumstances. This flexibility allowed participants to postpone the recording to a later date when they were feeling better and able to provide more representative dietary information.

Furthermore, at the conclusion of the dietary record, participants were queried about whether the recorded day was considered a "normal" or "classic" day in their life, or if any extraordinary circumstances, such as illness or unusual events, were encountered. This query served as a vital assessment to determine the overall representativeness of the recorded day. To mitigate potential biases associated with health-related disruptions during the recording of food records, we have incorporated this information as a confounder into a sensitivity analysis. The results of this adjustment are now presented in Reviewer Table 3 and in Supplementary material (Table s1, Model 16), and as observed, this additional information had minimal impact on our findings. We have added this information in the Methods (p.10, l.275): *“In the dietary records, participants were asked whether their 24-hour record was representative of a regular day or whether it was impacted by any extraordinary circumstances, such as illness or unusual events. We adjusted for this information.”* and results (p. 13, l. 369): *“The inclusion of number of medications (Model 14), unusual dietary reporting (Model 16) and restrictive diets (Model 17) did not impact these results.”*

5. Line 127. An eating occasion was defined as the intake of any food or beverage of at least 1 kcal. What about the intake of medication (that also delivers some calories, and can be taken with a caloric drink)? What about intake of noncaloric artificially sweetened beverages and chewing gum?

We thank the reviewer for this comment. In our study, the definition of an eating occasion was based on the intake of any food or beverage providing at least 1 kcal, with the intention of capturing all eating behaviours while excluding water intake. This conservative approach was adopted because there is currently no established evidence on the specific threshold of kcal required to synchronize and activate the peripheral clocks of the circadian system.

By defining an eating occasion as any intake of at least 1 kcal, we aimed to capture a wide range of eating behaviours, including situations where individuals may consume a small amount of food or a caloric/low-caloric beverage early in the morning before consuming a more substantial breakfast later on. This approach helps to minimize potential misclassification bias that could occur if individuals do not consider such early morning intake as a formal breakfast. With our

approach, we capture any eating occasion without a predefined definition. This was further explained in the methods section (p. 5, l. 133) as follows “*We have made this assumption as we wanted to be conservative by capturing any eating occasion without a predefined definition (including, for example, a coffee with milk or other caloric or low-caloric drinks).*”

Even though timing of medication was not collected, we have added a sensitivity analysis adjusting for medication use, as detailed in Question 2. This adjustment did not significantly alter our results (model 14, supplementary table 1).

Non-caloric artificially sweetened beverages were reported by participants in the food records but were not considered as separate eating occasions since they do not contribute significant calories. As for chewing gum, its consumption was not explicitly captured in the food records, and therefore, it was not considered in the definition of eating occasions in our study.

6. Line 129-130. The authors excluded participants reporting the consumption of a first meal after 3PM or a last meal before 3PM since they corresponded to very disrupted circadian behaviours. Is that always true? People having their last meal at 3PM and first meal at 7AM may be intermittent fasters. Why not included as a confounder in the analysis, instead of excluding these people? How were people with intermittent fasting identified/analysed? Were people on a weight loss diet or other prescribed diet excluded?

In our main analyses, we wanted to exclude these participants as we considered these behaviours to be a proxy of night shift, related to disrupted rhythms. However, in sensitivity analyses we explored including them back in the sample (Supplementary material, Table S1, model 12) and results remained unchanged compared to our main model.

At the end of each dietary record, participants were asked whether they were following any caloric restriction or a restrictive diet. This helps to identify energy under reporters from participants having plausible low energy intake because of restrictive diets or weight loss. “True” energy under reporters are excluded, but those who reported practicing caloric restriction were included back in the analyses. It was the case for 221 participants in our study population who had a dietary intake of less than 800kcal per day but were included in our analyses. This was explained in the Supplementary material, Method A as follows: “*Some under-reporting individuals were not excluded if their reported energy intake, initially estimated abnormally low, was found to be likely in case of recent weight variation or reported practice of weight-loss restrictive diet or proactive statement of the participant that he/she ate less than usual on the day of the dietary record.*”

In sensitivity analyses, we explored adjusting for a variable stating if participants were on a restrictive diet or not. We present these new results in Reviewer Table 4 and in supplementary material (Table S1 model 16). Results remained unchanged compared to those in table 2 in the manuscript.

We have added this info in methods (p. 10, l. 278): “*We also adjusted for the practice of caloric restriction or restrictive diet*” and results (p. 13, l. 369): “*The inclusion of number of medications (Model 14), unusual dietary reporting (Model 16) and restrictive diets (Model 17) did not impact these results*”

7. Line 131-133 The authors calculated nighttime fasting duration as follows: 24h minus the time elapsed between the first and the last meal of the day. —> intakes of

>1 kcal were considered eating episodes. So if people just had 1 candy (e.g. for sore throat) during the night, was that considered as an eating episode or disrupted fast?

We thank the reviewer for this comment. In the NutriNet-Santé cohort, participants were asked to report the consumption of any food or beverage during a 24h period. We assumed that participants reported eating occasions from the time they woke up until the time they went to bed. Therefore, in the situation mentioned by the Reviewer, the participant would have also reported any eating occasion occurring at 1am, as it occurred before them going to bed. If this intake was of 1kcal or more this would have been identified as the last eating occasion of the day. Past midnight eating occasions were therefore considered. Nevertheless, information bias linked to misreporting of past midnight eating occasions could not be entirely ruled out.

8. Line 153-155. Did the authors consider sleep apnea as CVD risk factor, and if so, what was the impact of this factor on the associations?

We agree with the reviewer that sleep apnea can affect CVD risk (Tietjens, JAHA 2018) and following this suggestion, we have now taken it into account. In the NutriNet-Santé study, a subset of the population responded to a comprehensive sleep questionnaire during follow-up (2014). This questionnaire included a question on sleep apnea and 428 participants reported having this condition. In sensitivity analyses, we have now added a new model adjusting for sleep apnea (Supplementary material, Table S1, model 11) and overall results remained unchanged meaning that the observed associations are independent of this sleep alteration. We have added this new information in the manuscript, p. 9, l. 266: *“We also explored adjusting for number of awakenings during the night and for sleep apnea as an indicator of sleep quality.”*

9. Line 161-162. Please show the distributions for eating times and frequency. Data are not normally distributed. Is it justified to analyse these data continuously?

In our study, the distributions of eating times and frequency were examined to understand the patterns and variations within the dataset. We acknowledge that these variables may not follow a normal distribution, as eating behaviours can vary widely among individuals and are influenced by various factors (Reviewer figure 2). We have also added this new figure in the supplementary material (Figure S3). It is important to note that normality is not an assumption that should be met in Cox models (Bradburn, BJC 2003), whereas the proportional hazards assumption is crucial to be assessed. This assumption implies that the hazard ratio (HR) remains constant over follow-up time.

In our analysis, we had assessed the proportional hazards assumption by plotting the Schoenfeld Residuals (Supplementary material, Figure S5) and ensured that it hold for the variables included in the Cox model. While the eating times and frequency may not follow a normal distribution, the Cox model remains a suitable approach for analysing time-to-event data, allowing us to examine the associations between these variables and the risk of CVD events.

Reviewer figure 2. Violin plots to show the distribution of our main exposure variables.

10. Line 169-170. How did the authors deal with unstable AP and TIA in their analyses of CHD and stroke, respectively?

Since we aimed to investigate the risk of the **first** cardiovascular event, we used a competing risks approach: To address the specific aim of investigating the risk of the first cardiovascular event, we used a competing risks approach in our analysis using cause-specific Cox models, as recommended for the studying etiologic associations in the presence of competing risks (Kragh, Int J Epi 2012).

In the context of our study, when studying cardiovascular diseases, competing risks refer to the presence of multiple possible outcomes (e.g., CHD and stroke) that may both occur independently, but compete with each other to be the first cardiovascular event.

In our models investigating risk of stroke, if a participant was diagnosed with a TIA during the follow-up, we applied censoring at the date of the TIA diagnosis, as per the cause-specific Cox model guidelines. This means that the participant was considered a non-case for stroke from that point onward, and stopped contributing to person-years. The same rationale was applied when studying CHD/myocardial infarction, if a participant was diagnosed with unstable AP during the follow-up, we applied censoring at the date of the AP diagnosis, and considered a non-case for CHD.

11. Line 180-196.

- a. Why adjusted for absolute alcohol intakes, rather than alcohol consumption patterns? The latter would be more relevant when studying meal timing. The same holds for physical activity.**

In our main analyses, we adjusted for both absolute alcohol intakes in grams/day (as a categorical variable, to account for non-linearity), which is an established risk factor for cardiovascular disease (Hoek, Curr Atheroscler Rep 2022), but also for number of episodes of binge drinking. Binge drinking is a behaviour typically associated with consuming a large amount of alcohol in a short period, often occurring at night in social settings or party environments. By including binge drinking as a covariate in our analyses, we indirectly accounted for the timing aspect of alcohol

consumption. Participants with frequent occurrences of binge drinking may be more likely to consume alcohol later in the day.

Regarding physical activity, and as stated above, we acknowledge the relevance of considering activity patterns in relation to meal timing. However, in our study, we did not have detailed data on the timing of physical activity. We appreciate the reviewer's insightful comment, and we have provided further clarification in the revised manuscript to address these limitations and potential for future investigations in the discussion. Now this section of the discussion (p. 17, l. 517) reads as follows: "*We had no information on exposure to light-at-night, use of recreational drugs, as well as timing of physical activity, medication or alcohol consumption which are all potential disruptors of circadian rhythms. Future studies collecting accelerometry data could be of interest to objectively account for physical activity timing and patterns.*"

b. Were sleep duration and quality, and number of awakening episodes measured? Do the authors consider these factors as confounders or intermediary factors? Could they include these variables in their models for people who had these data?

We thank the reviewer for this comment. In the NutriNet, a subset of the study population responded to an optional comprehensive sleep questionnaire during follow-up (2014). From this questionnaire we calculated sleep duration (hrs. / 24h, continuous) and time interval between bedtime and last meal. Then, in sensitivity analyses we explored adjusting our models for these variables (Table S1, model 9). Results remained robust.

As suggested by the Reviewer, we have now added a variable about the number of awakenings as a proxy for sleep quality in a sensitivity analysis (Table S1 model 11), and we have added the following in the text (p.9, l. 265): "*We also explored adjusting for number of awakenings during the night and for sleep apnea as an indicator of sleep quality*". We also added this information in the results (p. 12, l. 365): "*Adjusting for sleep quality indicators (Model 11), for marital status and number of children (Model 13) did not change the observed associations.*"

c. BMI is considered a confounder, and is part of all multivariable models. However, BMI could also be an intermediary factor, as well as an effect modifier. Would it not be better to stratify for BMI and/or to include it in sensitivity analysis?

We acknowledge that the relationship between BMI and meal timing is complex. As raised in Reviewer 1's specific question (Question 1), there is a possibility that individuals with overweight at baseline might skip breakfast as a weight management strategy (as depicted in Reviewer Figure 3). To address this potential confounding effect, we incorporated BMI at baseline as an adjustment in our primary models. Moreover, we explored the notion that the association between meal timing and CVD risk might be mediated by weight gain (as illustrated in Reviewer Figure 3). In response, we conducted sensitivity analyses, adjusting for weight change per year during follow-up (detailed in Table S1, Model 3). Weight change was computed as the percentage of weight change from baseline until the end of follow-up, divided by the number of years of follow-up. The results from these analyses remained largely unchanged, implying that the indirect mediated effect through weight change is minor, and that other mechanisms may underlie the observed associations. Furthermore, we investigated the possibility of BMI acting as an effect modifier (explained in Methods, page 8, line 225). However, the p-value for interaction was not statistically significant, and the estimates across strata were similar.

Reviewer figure 3. DAG representing the associations between baseline BMI, meal timing variables, BMI change and CVD risk.

BMI t_0 representing BMI at baseline; BMI t_1 representing body weight change.

d. Was the time between awakening and first meal recorded? If so, should that variable not be included in the model?

We made the decision not to include the time between awakening and first meal in the main models due to the lack of evidence, and limitations in data availability.

Firstly, to our knowledge, unlike for the difference between the last eating episode and bedtime (Kogevinas, IJC 2018), there is no existing evidence in the literature suggesting a significant association of the time between awakening and first meal with cardiovascular or human health. While there are studies that explore the relationship between meal timing and health outcomes, the specific variable of time between awakening and first meal is not a well-established predictor of cardiovascular health.

Secondly, the assessment of sleep, including the time of awakening, was recorded in a questionnaire later in time and only for a subset of the study population. Although we used these variables for sensitivity analyses, these are not ideal for the main models. This limited availability of data may introduce selection bias and decrease the generalizability of the findings.

e. Were data collected on marital status? What is the impact of being married/single/widowed etc and having children on eating patterns and risk of CVD? Social factors (adaptation to household members) could play a major role in meal patterns, and are also related to CVD risk.

We thank the reviewer for raising this interesting point. In the NutriNet-Santé cohort this information was collected at baseline and we now adjust for both marital status (Married; Couple; Divorced or separated; Widower; Single) and number of children. We have added the corresponding text in the manuscript and present these new results in supplementary material (Table S1, Model 13). As we can observe, adjusting for these variables did not importantly change the results for the association between time of first meal and risk of overall cardiovascular diseases nor between time of last meal and risk of cerebrovascular diseases. We have added this information in the methods section (p. 10, l. 269): “Additionally, we explored adjusting our models for marital status and number of children.” We also added this information in the results (p. 12, l. 365): “Adjusting for sleep quality indicators, for marital status and number of children did not change the observed associations.”

12. Line 245-260. Did the authors consider sensitivity analyses after exclusion of (cardiovascular, psychotropic or other) drug users? Idem, after exclusion of people

with disturbed sleeping patterns and/or sleep apnea? Idem, after exclusion of people with (morbid) obesity or diabetes?

We did not try excluding participants using cardiovascular nor psychotropic drugs, however, in reviewer table 1 (and table S1 model 14) we adjusted for number of medications as a proxy and results remained unchanged. Furthermore, we can assume that use of cardiovascular medication at baseline was not very frequent, as participants with a prevalent cardiovascular condition were excluded from the sample. In our main model we did adjust for baseline BMI and in sensitivity analyses we adjusted for weight change during follow-up (Table S1, model 3) and for prevalent diabetes (Table S1, model 6). As suggested by the Reviewer, we now additionally explored excluding prevalent cases of obesity and diabetes. Similarly, we have examined the exclusion of those that in the sleep questionnaire reported having apnea. We present these new results in Reviewer table 5 and supplementary table S2. As we can see these results remained unchanged even after the exclusion of these groups at higher risk of CVDs.

We added this new information in methods (p.10, l. 279): *“Moreover, we performed models where we excluded from the sample participants with prevalent obesity, type 2 diabetes, or sleep apnea”* and results (p. 13, l. 371): *“Finally, the exclusion of prevalent cases of obesity, type 2 diabetes and apnea did not change the observed associations (Table S2).”*

13. Table 1

- a. People with lower education tend to have their breakfasts earlier in the morning. The opposite is true for income: those with lower incomes have their breakfasts later. I would not expect that. Have data been collected on type of job? What about unemployment or sickness? It is well-known that people with a lower socio-economic position (SEP) have a higher risk of CVD. Could residual confounding have impacted the conclusion of this study? Since SEP is one of the most important confounding factors in this study, the authors should further explore this (e.g. in sensitivity analyses).**

In the NutriNet-Santé cohort, we collected data on several relevant socioeconomic indicators to account for their potential confounding effects. Specifically, we adjusted all our models for educational level, which was categorized as "less than high school degree," "<2 years after high school degree," and "≥2 years after high school degree." Additionally, we accounted for monthly income per unit of consumption using categories such as "less than 900€," "900-1,200€," "1,200-1,800€," "1,800-2,300€," "2,300-3,700€," "more than 3,700€," and "don't want to answer."

Furthermore, we conducted sensitivity analyses to explore the impact of profession on our results. In the Supplementary material (Table S1, Model 2), we presented a model that was further adjusted for profession, with categories including "Unemployed," "Student," "Self-employed, farmer," "Employed, manual worker," "Intermediate professions," "Managerial staff," "Intellectual profession," and "Retired." We found that the results did not significantly change compared to our main model, indicating that the confounding effect of profession is rather minor. We have now included this variable in table 1 to see the distribution of professions across groups. Even though the trend highlighted by the Reviewer might seem odd, it could be explained by differences in age and generation: indeed, participants reporting later first meals were more likely to be students and younger in our study (which explains the lower income), but also to have higher educational levels, which is more common among younger generations; while those reporting earlier first meals were more likely to be older, which is consistent with them having a higher income.

While we acknowledge that residual confounding is always a possibility in observational studies, we made concerted efforts to account for relevant socioeconomic factors in our analysis.

b. Include information on marital status, medication use and sleep parameters. Consider these factors in the analysis.

We thank the reviewer for this suggestion. We have now added marital status, number of used medications at baseline, and sleep parameters (for the subpopulation responding to the sleep questionnaire) in table 1. As we can see in table 1, number of medications was similar across groups. We have also added this new information in p. 10, l. 294, as follows: “*Overall, younger participants, students or unemployed, single, without a family history of CVD, current regular smokers, with higher physical activity levels, higher educational levels and lower monthly incomes tended to have later first and last meals. Additionally, compared to participants with earlier meals, participants having later meals had a higher consumption of alcohol, more episodes of binge drinking, reported later bedtimes and they were more likely to have a higher variability in their meal timings across the week (defined as eating jet lag).*”

c. Why are men underrepresented in the analyses? How is that related to job activities, taking care of household, children etc.? Please explain and consider potential selection bias in the results.

In the NutriNet-Santé cohort, like many other volunteer-based cohorts, there is a higher proportion of women compared to men (Andreeva, J Epidemiol Community Health 2015). This disparity is primarily due to differences in participation rates, as women tend to be more self-conscious about their health and are more likely to actively engage in research studies, particularly related to nutrition. The underrepresentation of men in our analyses is a reflection of this enrolment pattern.

Gender is surely related to job activities such as job responsibilities, household duties, and childcare. These factors may contribute to differences in lifestyle behaviours and meal timing. Sex and profession were incorporated in our analyses (sex is adjusted for in all models and sex-specific results are presented, profession was adjusted for in supplementary material, Table S1 Model 2). We have also now incorporated a model adjusting for marital status and number of children (Table S1, Model 13) and present these new results in the revised version of our manuscript. These new results have been described in detail in question 11e of Reviewer 1.

One of the main objectives of the NutriNet-Santé cohort is to investigate the relationship between diet and health. NutriNet-Santé is a web-based cohort, and participants are recruited on a voluntary basis. Therefore, NutriNet-Santé does not aim to be representative of the general French population, but instead to have enough contrast between different categories of dietary behaviours to conduct etiological analyses, while accounting for a wide diversity of lifestyle profiles. As it is usually the case in population-based cohorts, participants in NutriNet-Santé were younger, more often women, and had higher socio-professional and educational levels than the general (French) population (Andreeva, J Epidemiol Community Health 2015).

Nevertheless, we cannot rule out the possibility of selection bias, as we discuss in the limitations section (p. 18, l. 538) as follows: “*Participants in the NutriNet-Santé cohort are volunteers and are more likely to be women, have a higher socioeconomic status and healthier behaviour patterns than the general population, somehow limiting the extrapolation of these results*⁸⁰. *Moreover, the healthier behaviours in the study population could have led to a lower incidence of CVD compared to the general population and to an underestimation of the studied associations,*

even though an overestimation cannot be totally excluded, as women, overrepresented in our study, tended to have later times of first meal than men.”

14. Table 4: Because of effect modification, sex-specific analyses as presented in Table 4 should be considered as primary analyses rather than the total cohort. The number of men is less than the number of women, but still sufficiently large for sex-specific analyses.

While we understand the merits of presenting sex-specific results first when effect modification is evident, we chose to show the results for the total population as the primary analysis to provide a comprehensive overview of the overall relationship between meal timing and CVD risk in the entire cohort before delving into sex-specific differences.

We believe that presenting the overall results first allows readers to grasp the general trends and associations before going into potential sex-specific nuances. In addition, this would be of better relevance for readers among cohort participants.

Even though the sample is sufficiently large for sex-specific analyses, sensitivity analyses within the sleep subsample would be challenging to conduct among men only.

We have clarified this in the methods section (p. 8, l. 231): *“Despite significant interactions with sex, overall results were presented first to provide an overview of the general trends, before presenting sex-specific results, for readability purposes.”*

15. Discuss whether awakening occurred naturally or otherwise, by alarm clock or children waking up parents. If no data are available on these factors, discuss the potential impact based on the literature.

Unfortunately, we did not collect specific data on the method of awakening (natural, or alarm clock or children waking up parents) in our study. Therefore, we were unable to incorporate these factors into our analysis.

However, we did consider the potential influence of having children on meal timings and its association with cardiovascular disease (CVD) risk. As mentioned in previous responses (see 11e), we adjusted our analyses for the number of children as a proxy for the presence of family responsibilities, including potential disruptions in sleep and meal patterns. The association between time of first meal and overall CVD risk and between time of last meal and risk of cerebrovascular diseases remained unchanged. These adjustments may not capture all nuances related to awakening methods or the influence of children on meal timing. We have now added this as a limitation in the discussion (p. 17, l. 522), that now reads:

“Furthermore, potential measurement imperfections in collected data despite adjustment (e.g., job-related factors, season, deliberate choice of meal timing and awakening, partying) as well as other unknown or unmeasured potential confounders could be contributing to residual confounding in this study. For instance, data about natural vs. alarm clock awakening or children waking their parents up were not available in our cohort, and might have introduced unmeasured confounding, since the method of awakening could influence sleep inertia and quality, and could therefore have an impact on hunger and meal timing^{83,84}.”

16. Elaborate on the role of intermittent fasting and the results from this study. It is not yet clear to me whether abstaining from food (and/or beverages) after dinner until breakfast is recommended or not, and how long that fasting period should last.

Periods of 12 hours or even 16 hours are recommended. For a country like the Netherlands, the latter would imply a fasting period from e.g. 7PM to 11AM. Do the authors consider that unhealthy, and why?

We thank the reviewer for this comment and we agree that these results are complex, as solid evidence is not yet available.

Our findings suggest that delaying the first and the last meals were both associated, independently, with a higher risk of cardiovascular outcomes, particularly among women. These findings, along with the broader evidence on breakfast skipping (Ofori-Asenso, *J Cardiovasc Dev Dis* 2019) highlight the importance of considering the potential negative aspects of delaying the first meal. Our analyses also suggest a potential protective role of a longer nighttime fasting duration on cerebrovascular diseases. Taken together with the results on the time of first and last meals, these findings are suggestive of a potential protective association between a longer nighttime fasting duration (i.e., time-restricted eating) and cardiovascular health, only if coupled with early first and last meals, rather than by skipping breakfast. Indeed, when considering fasting periods, it is also crucial to take into account the time when fasting occurs during the day. For instance, fasting for 14 hours starting at 6 PM and ending at 8 AM may have different implications compared to fasting from 10 PM to 12 PM.

Although we explored the potential effect modification of the association between nighttime fasting duration and cardiovascular diseases by the time of the first meal of the day, the results were not statistically significant. However, it is worth noting that our previous results from the NutriNet-Santé cohort suggest that nighttime fasting duration was not associated with type 2 diabetes incidence, except in participants having breakfast before 8AM and fasting for >13 h overnight (Palomar-Cros, *Int J Epi* 2023). In another investigation conducted among participants of the Spanish multicase-control study (Palomar-Cros, *Nutrients* 2021), we found that the association between nighttime fasting duration and prostate cancer risk was modified by the time of the first meal. Among participants who had breakfast at 8:30 AM or earlier, a longer fasting period was associated with a lower risk of prostate cancer, consistently with the results on type 2 diabetes in NutriNet-Santé.

Overall, the emerging evidence suggests that a longer fasting period overnight may have potential health benefits as long as it is accompanied by an early last eating occasion rather than skipping breakfast (i.e., early time-restricted eating). This is an important message to consider, as some fasting trends promote the idea of skipping breakfast and delaying the first meal until lunch. Our findings suggest that delaying the first meal could be associated with negative health outcomes. This is now thoroughly explained in the discussion (p. 16, l. 465), that now reads:

“Noticeably, the time of the day when the fasting period occurs could also play a role in cardiometabolic health. Considering evidence on the negative impacts of delaying the first and the last meal of the day, it is reasonable to think that it would be better to practice TRE by having an early first and last meal of the day, i.e., early-TRE (eTRE). Recent interventional studies are starting to demonstrate that eTRE is more beneficial than other fasting regimens with later eating windows throughout the day 80. Moreover, this could explain the discrepant results in some studies on TRE with later eating schemes (eating window from 12PM to 8PM²⁵ or from 10AM to 7PM²⁴). Interestingly, a study among participants of the NHANES survey showed that in women, extending the nighttime fasting duration was associated with lower levels of C-reactive protein level (but not with insulin resistance models) only among those who ate fewer than 30% of their calories after 5PM⁷⁷. Although in our study population we did not find a statistically significant

interaction between nighttime fasting and the time of the first meal of the day, our results on nighttime fasting duration, taken together with those on the time of first and last meals, are suggestive of a potential protective association between a longer nighttime fasting duration (i.e., TRE) and cardiovascular health, only if coupled with early first and last meals (i.e., eTRE), rather than by skipping breakfast. These findings are somehow in line with our previous results from the NutriNet-Santé cohort suggesting that nighttime fasting duration is only associated with type 2 diabetes incidence in participants having breakfast before 8AM and fasting for >13 h overnight⁸¹. In another investigation conducted among participants of the Spanish multicase-control study⁸², we also found that the association between nighttime fasting duration and prostate cancer risk was modified by the time of the first meal. Among participants who had breakfast at 8:30 AM or earlier, a longer fasting period was associated with a lower risk of prostate cancer.”

17. Discuss residual confounding more thoroughly. See general comments.

- **Residual confounding by socioeconomic status (e.g. job-related factors) and/or unemployment or lack of daytime activities**
- **Residual confounding by marital status and household factors (are meal times chosen deliberately, or do people have to wake up at certain times because of children, jobs, studies?)**
- **The role of intermittent fasting, slimming diets, and other prescribed diets, which is not accounted for in the analyses**
- **The role of medication (type, dose, timing, with meals or in-between)**
- **Reverse causation bias, e.g. people with poor health who have difficulty getting out of bed in the mornings because of health problems, poor sleep quality etc.**
- **Residual confounding by behavioral factors that make people stay up till late, e.g. partying, alcohol intake pattern and the independent role these factors have on CVD risk**
- **Residual confounding by season, which affect circadian rhythms, sleeping times, meal patterns, and also risk of CVD.**

We have discussed each of these points individually in the answers above. However, we agree that the discussion lacked more caveats on residual confounding. Now it reads as follows (p. 17, l. 516): *“Although we accounted for a large panel of confounders, given the observational nature of this study, residual confounding cannot be completely ruled out. We had no information on exposure to light-at-night, use of recreational drugs, as well as timing of physical activity, medication or alcohol consumption which are all potential disruptors of circadian rhythms. Future studies collecting accelerometry data could be of interest to objectively account for physical activity timing and patterns. Furthermore, potential measurement imperfections in collected data despite adjustment (e.g., job-related factors, season, deliberate choice of meal timing and awakening, partying, chronotype) as well as other unknown or unmeasured potential confounders could be contributing to residual confounding in this study. For instance, data about natural vs. alarm clock awakening or children waking their parents up were not available in our cohort, and might have introduced unmeasured confounding, since the method of awakening could influence sleep inertia and quality, and could therefore have an impact on hunger and meal timing. We used a falsification endpoint that is not expected to be related to the exposure in a way to reduce the risk of spurious findings in our main models. In these analyses, our exposures of interest were not associated with the negative control (basal cell carcinoma). Moreover, reverse causation bias linked to change of behaviours in participants with poor health having difficulty*

getting out of bed in the mornings because of health problems, cannot be entirely ruled out, despite efforts made in the sensitivity analyses.”

18. Line 398-400. The authors state that “The association between late first and last meals and a higher overall CVD risk was stronger in women than in men. These differences could be mainly explained by the higher proportion of women in our cohort.” This reasoning is not correct. Sample size has an impact on precision (confidence intervals), not on the measure of association (relative risks).

The reviewer is right. We apologise. We agree that the way it was written this was confusing and have deleted the second part of the sentence.

19. Line 408 If BMI could be part of the biological pathway for meal timing in relation to CVD, why did the authors adjust for it in their multivariable models?

We thank the reviewer for this comment. BMI at baseline can act as a confounder in our study (see Reviewer figure 3). As BMI reflects the body mass index at the beginning of the study, it may capture potential effects of past dietary behaviours, physical activity, and lifestyle factors on both meal timing preferences and CVD risk. Failing to adjust for BMI in our analysis could lead to biased estimates of the association between meal timing and CVD outcomes, as the observed relationship might be confounded by baseline BMI.

On the other hand, we agree that BMI change over the study period could be a potential mediator in the biological pathway between meal timing and CVD risk (see Reviewer figure 3). We have approached this by adjusting for weight change during follow-up in sensitivity analyses (Table S1, Model 3). Results did not substantially change, suggesting other mechanisms could be explaining these associations.

Reviewer #2 (Remarks to the Author):

This is well written manuscript that examines how meal timing and nighttime fasting are related to cardiovascular health parameters. The topic is novel and will be of interest to the readers of this journal. The methods are sound, and the results are not overstated in the conclusion. I only have a few minor comments:

We thank Reviewer #2 for the very helpful and constructive comments.

1. Chronotype was not measured in the prospective. It would have been important for chronotype to be recorded for each subject by the morningness-eveningness questionnaire (MEQ). Please comment on how not using the MEQ may have limited your findings.

We thank the Reviewer for bringing up this important point. In this prospective study, chronotype was indeed measured in a subset of participants (N=44,591) during the follow-up in 2014. We apologize for not providing this information clearly in the initial manuscript, and we have now updated the Methods section (p. 5, l. 108) to include more details on how chronotype was assessed. It now reads as follows: *“In 2014, a subset of the study population responded to an optional comprehensive sleep questionnaire, which included a question on chronotype (Do you consider yourself as: clearly a morning person, rather a morning person, clearly an evening person, rather an evening person, neither morning or evening, both or I don’t know) ³².”*

This approach allowed us to capture a broad understanding of participants' chronotype preferences; however, we understand that it may not provide the same level of precision as more comprehensive measures such as the Morningness-Eveningness Questionnaire (MEQ).

Chronotype has indeed been associated with increased CVD risk (Baldanzi, Sleep 2022), and might therefore be a confounder that should be better measured. We have expanded the discussion section to address the limitations of our approach to measuring chronotype and the potential impact it may have had on our findings. P. 17, l. 510: “*Similarly, chronotype was assessed in this same questionnaire and it was self-reported by the participants, however, specific questionnaires such as the morningness - eveningness questionnaire (MEQ) could have been more appropriate tools.*”

2. Line 351 **✎ Please also comment on the BMJ meta-analysis that reports no negative effect of breakfast skipping on body weight (<https://www.bmj.com/content/364/bmj.142>) As it stands, this sentence is only reporting the negative outcomes of breakfast skipping, and lacks equipoise.**

We have added the study suggested by the reviewer as follows (p. 13, l. 385): “*However, a recent meta-analysis found no impact of breakfast skipping on body weight*⁵⁸.”

3. In the discussion, please comment on why some countries (e.g., Spain, Argentina, Chile, etc) that typically eat dinner late in the evening (10-11pm) do not have higher risk of CVD compared to countries (USA, Canada, France, etc) that typically eat dinner earlier in the evening (6-8 pm).

We thank the Reviewer for this comment. While it is true that certain countries, such as Spain, Argentina, Chile, and others, have cultural practices of eating dinner later in the evening (around 9-11 pm) compared to countries like the USA, Canada, France, Germany, where dinner is typically consumed earlier (around 6-8 pm), it is important to consider, when using these ecological approaches, multiple factors that contribute to the overall risk of CVD in these populations.

Meal timing alone is not the sole determinant of CVD risk disparities between these countries. Other demographic, lifestyle and dietary factors, including smoking and alcohol use, the types of foods consumed, portion sizes, overall dietary patterns, physical activity levels, socioeconomic factors, and genetic predispositions, could also play significant roles, as well as environmental factors.

We have added some elements to the discussion (p. 19, l. 553), as follows: “*Lastly, even though meal timing showed associations with cardiovascular outcomes in our study, CVD remains a multifactorial disease, and meal timing alone could not explain the trends in CVD incidence across countries, as other demographic, lifestyle, genetic and environmental factors are involved.*”

Reviewer table 1. Association of meal timing and number of eating occasions with risk of cardiovascular diseases in the NutriNet-santé cohort, 2009-2021, N=103,389. Additionally adjusting for medication use.

	N cases / non-cases	HR (95% CI) ¹	p-val	N cases / non-cases	HR (95% CI) ²	p-val
Overall cardiovascular diseases						
Time of first meal (1h incr.)	2,036 / 101,353	1.06 (1.01 – 1.12)	0.02	2,036 / 101,353	1.05 (1.00 – 1.10)	0.06
Time of last meal (1h incr.)	2,036 / 101,353	1.02 (0.98 – 1.07)	0.4	2,036 / 101,353	1.02 (0.97 – 1.06)	0.4
Number of eating occasions (1 occasion incr.)	2,036 / 101,353	0.99 (0.96 – 1.02)	0.5	2,036 / 101,353	0.98 (0.95 – 1.01)	0.2
Cerebrovascular diseases ‡						
Time of first meal	988 / 102,401	1.06 (0.98 – 1.14)	0.1	988 / 102,401	1.05 (0.97 – 1.12)	0.2
Time of last meal	988 / 102,401	1.08 (1.01 – 1.15)	0.02	988 / 102,401	1.07 (1.01 – 1.15)	0.03
Number of eating occasions	988 / 102,401	0.97 (0.93 – 1.01)	0.1	988 / 102,401	0.96 (0.92 – 1.00)	0.08
Coronary heart diseases §						
Time of first meal	1,071 / 102,318	1.05 (0.98 – 1.13)	0.1	1,071 / 102,318	1.04 (0.97 – 1.11)	0.3
Time of last meal	1,071 / 102,318	0.97 (0.92 – 1.05)	0.4	1,071 / 102,318	0.97 (0.91 – 1.03)	0.4
Number of eating occasions	1,071 / 102,318	1.01 (0.97 – 1.05)	0.6	1,071 / 102,318	1.00 (0.96 – 1.04)	0.9

HR= Hazard ratio; N= Sample size; CI= Confidence Interval.

‡ Stroke and transient ischemic attack. § Myocardial infarction, acute coronary syndrome, angioplasty and angina pectoris.

1. Main model in Table 2 (manuscript). Multivariable Cox proportional hazard models adjusted for age (timescale), sex (women, men), educational level (less than high school degree, <2 years after high school degree, ≥2 years after high school degree), monthly income per unit of consumption (less than 900€, 900-1,200€, 1,200-1,800€, 1,800-2,300€, 2,300-3,700€, more than 3,700€, don't want to answer), BMI at baseline (continuous, kg/m²), family history of CVDs (no, yes), alcohol consumption (Non-consumers (0g/day), low consumers (0.1 – 4.9 g/day), moderate consumers (5.0 – 14.9 g/day), high consumers (15.0 -29.9 g/day) and very high consumers (>30.0 g/day)), episodes of binge drinking (None, one, more than one), daily energy intake excluding alcohol (continuous, kcal/day), healthy and Western dietary patterns derived by factorial analysis (continuous), smoking (current regular (1 cigarette or more per day), current occasional, former, never), number of pack years (continuous, defined as the number of packs of cigarettes smoked per day by the number of years of smoking), physical activity (low, moderate, high), number of dietary records (continuous). Time of first and last meal and number of eating occasions were mutually adjusted. 2. Adding **number of medications**.

Reviewer table 2. Association of meal timing and number of eating occasions with risk of cardiovascular diseases in the NutriNet-santé cohort, 2009-2021, N= 103,389. Additionally adjusting for season when the questionnaire of physical activity was completed.

	N cases / non-cases	HR (95% CI) ¹	p-val	N cases / non-cases	HR (95% CI) ²	p-val
Overall cardiovascular diseases						
Time of first meal (1h incr.)	2,036 / 101,353	1.06 (1.01 – 1.12)	0.02	2,036 / 101,353	1.06 (1.01 – 1.12)	0.01
Time of last meal (1h incr.)	2,036 / 101,353	1.02 (0.98 – 1.07)	0.4	2,036 / 101,353	1.02 (0.97 – 1.07)	0.4
Number of eating occasions (1 occasion incr.)	2,036 / 101,353	0.99 (0.96 – 1.02)	0.5	2,036 / 101,353	0.99 (0.96 – 1.02)	0.5
Cerebrovascular diseases ‡						
Time of first meal	988 / 102,401	1.06 (0.98 – 1.14)	0.1	988 / 102,401	1.06 (0.99 – 1.14)	0.1
Time of last meal	988 / 102,401	1.08 (1.01 – 1.15)	0.02	988 / 102,401	1.08 (1.01 – 1.15)	0.03
Number of eating occasions	988 / 102,401	0.97 (0.93 – 1.01)	0.1	988 / 102,401	0.97 (0.93 – 1.01)	0.1
Coronary heart diseases §						
Time of first meal	1,071 / 102,318	1.05 (0.98 – 1.13)	0.1	1,071 / 102,318	1.05 (0.98 – 1.13)	0.1
Time of last meal	1,071 / 102,318	0.97 (0.92 – 1.05)	0.4	1,071 / 102,318	0.97 (0.92 – 1.04)	0.4
Number of eating occasions	1,071 / 102,318	1.01 (0.97 – 1.05)	0.6	1,071 / 102,318	1.01 (0.97 – 1.05)	0.6

HR= Hazard ratio; N= Sample size; CI= Confidence Interval.

‡ Stroke and transient ischemic attack. § Myocardial infarction, acute coronary syndrome, angioplasty and angina pectoris.

1. Main model in Table 2 (manuscript). Multivariable Cox proportional hazard models adjusted for age (timescale), sex (women, men), educational level (less than high school degree, <2 years after high school degree, ≥2 years after high school degree), monthly income per unit of consumption (less than 900€, 900-1,200€, 1,200-1,800€, 1,800-2,300€, 2,300-3,700€, more than 3,700€, don't want to answer), BMI at baseline (continuous, kg/m²), family history of CVDs (no, yes), alcohol consumption (Non-consumers (0g/day), low consumers (0.1 – 4.9 g/day), moderate consumers (5.0 – 14.9 g/day), high consumers (15.0 -29.9 g/day) and very high consumers (>30.0 g/day)), episodes of binge drinking (None, one, more than one), daily energy intake excluding alcohol (continuous, kcal/day), healthy and Western dietary patterns derived by factorial analysis (continuous), smoking (current regular (1 cigarette or more per day), current occasional, former, never), number of pack years (continuous, defined as the number of packs of cigarettes smoked per day by the number of years of smoking), physical activity (low, moderate, high), number of dietary records (continuous). Time of first and last meal and number of eating occasions were mutually adjusted. 2. Adding **season when the questionnaire on physical activity was completed.**

Reviewer table 3. Association of meal timing and number of eating occasions with risk of cardiovascular diseases in the NutriNet-santé cohort, 2009-2021, N= 103,389. Additionally adjusting for unusual dietary reporting.

	N cases / non-cases	HR (95% CI) ¹	p-val	N cases / non-cases	HR (95% CI) ²	p-val
Overall cardiovascular diseases						
Time of first meal (1h incr.)	2,036 / 101,353	1.06 (1.01 – 1.12)	0.02	2,036 / 101,353	1.06 (1.01 – 1.12)	0.01
Time of last meal (1h incr.)	2,036 / 101,353	1.02 (0.98 – 1.07)	0.4	2,036 / 101,353	1.02 (0.97 – 1.07)	0.4
Number of eating occasions (1 occasion incr.)	2,036 / 101,353	0.99 (0.96 – 1.02)	0.5	2,036 / 101,353	0.99 (0.96 – 1.01)	0.3
Cerebrovascular diseases ‡						
Time of first meal	988 / 102,401	1.06 (0.98 – 1.14)	0.1	988 / 102,401	1.06 (0.98 – 1.14)	0.1
Time of last meal	988 / 102,401	1.08 (1.01 – 1.15)	0.02	988 / 102,401	1.08 (1.01 – 1.15)	0.02
Number of eating occasions	988 / 102,401	0.97 (0.93 – 1.01)	0.1	988 / 102,401	0.97 (0.93 – 1.01)	0.1
Coronary heart diseases §						
Time of first meal	1,071 / 102,318	1.05 (0.98 – 1.13)	0.1	1,071 / 102,318	1.06 (0.99 – 1.13)	0.1
Time of last meal	1,071 / 102,318	0.97 (0.92 – 1.05)	0.4	1,071 / 102,318	0.97 (0.92 – 1.03)	0.4
Number of eating occasions	1,071 / 102,318	1.01 (0.97 – 1.05)	0.6	1,071 / 102,318	1.00 (0.96 – 1.04)	0.8

HR= Hazard ratio; N= Sample size; CI= Confidence Interval.

‡ Stroke and transient ischemic attack. § Myocardial infarction, acute coronary syndrome, angioplasty and angina pectoris.

1. Main model in Table 2 (manuscript). Multivariable Cox proportional hazard models adjusted for age (timescale), sex (women, men), educational level (less than high school degree, <2 years after high school degree, ≥2 years after high school degree), monthly income per unit of consumption (less than 900€, 900-1,200€, 1,200-1,800€, 1,800-2,300€, 2,300-3,700€, more than 3,700€, don't want to answer), BMI at baseline (continuous, kg/m²), family history of CVDs (no, yes), alcohol consumption (Non-consumers (0g/day), low consumers (0.1 – 4.9 g/day), moderate consumers (5.0 – 14.9 g/day), high consumers (15.0 -29.9 g/day) and very high consumers (>30.0 g/day)), episodes of binge drinking (None, one, more than one), daily energy intake excluding alcohol (continuous, kcal/day), healthy and Western dietary patterns derived by factorial analysis (continuous), smoking (current regular (1 cigarette or more per day), current occasional, former, never), number of pack years (continuous, defined as the number of packs of cigarettes smoked per day by the number of years of smoking), physical activity (low, moderate, high), number of dietary records (continuous). Time of first and last meal and number of eating occasions were mutually adjusted. 2. Adding **unusual dietary reporting (yes, no)**.

Reviewer table 4. Association of meal timing and number of eating occasions with risk of cardiovascular diseases in the NutriNet-santé cohort, 2009-2021, N= 103,389. Additionally adjusting for restrictive diet.

	N cases / non-cases	HR (95% CI) ¹	p-val	N cases / non-cases	HR (95% CI) ²	p-val
Overall cardiovascular diseases						
Time of first meal (1h incr.)	2,036 / 101,353	1.06 (1.01 – 1.12)	0.02	2,036 / 101,353	1.06 (1.01 – 1.12)	0.01
Time of last meal (1h incr.)	2,036 / 101,353	1.02 (0.98 – 1.07)	0.4	2,036 / 101,353	1.02 (0.97 – 1.07)	0.4
Number of eating occasions (1 occasion incr.)	2,036 / 101,353	0.99 (0.96 – 1.02)	0.5	2,036 / 101,353	0.99 (0.96 – 1.01)	0.3
Cerebrovascular diseases ‡						
Time of first meal	988 / 102,401	1.06 (0.98 – 1.14)	0.1	988 / 102,401	1.06 (0.99 – 1.14)	0.1
Time of last meal	988 / 102,401	1.08 (1.01 – 1.15)	0.02	988 / 102,401	1.08 (1.01 – 1.15)	0.02
Number of eating occasions	988 / 102,401	0.97 (0.93 – 1.01)	0.1	988 / 102,401	0.97 (0.93 – 1.01)	0.5
Coronary heart diseases §						
Time of first meal	1,071 / 102,318	1.05 (0.98 – 1.13)	0.1	1,071 / 102,318	1.05 (0.98 – 1.13)	0.1
Time of last meal	1,071 / 102,318	0.97 (0.92 – 1.05)	0.4	1,071 / 102,318	0.97 (0.92 – 1.04)	0.4
Number of eating occasions	1,071 / 102,318	1.01 (0.97 – 1.05)	0.6	1,071 / 102,318	1.00 (0.97 – 1.04)	0.8

HR= Hazard ratio; N= Sample size; CI= Confidence Interval.

‡ Stroke and transient ischemic attack. § Myocardial infarction, acute coronary syndrome, angioplasty and angina pectoris.

1. Main model in Table 2 (manuscript). Multivariable Cox proportional hazard models adjusted for age (timescale), sex (women, men), educational level (less than high school degree, <2 years after high school degree, ≥2 years after high school degree), monthly income per unit of consumption (less than 900€, 900-1,200€, 1,200-1,800€, 1,800-2,300€, 2,300-3,700€, more than 3,700€, don't want to answer), BMI at baseline (continuous, kg/m²), family history of CVDs (no, yes), alcohol consumption (Non-consumers (0g/day), low consumers (0.1 – 4.9 g/day), moderate consumers (5.0 – 14.9 g/day), high consumers (15.0 -29.9 g/day) and very high consumers (>30.0 g/day)), episodes of binge drinking (None, one, more than one), daily energy intake excluding alcohol (continuous, kcal/day), healthy and Western dietary patterns derived by factorial analysis (continuous), smoking (current regular (1 cigarette or more per day), current occasional, former, never), number of pack years (continuous, defined as the number of packs of cigarettes smoked per day by the number of years of smoking), physical activity (low, moderate, high), number of dietary records (continuous). Time of first and last meal and number of eating occasions were mutually adjusted. 2. Adding **restrictive diet (yes, no)**.

Reviewer table 5. Association of meal timing and number of eating occasions with risk of cardiovascular diseases in the NutriNet-santé cohort, 2009-2021, N= 103,368

	N cases / non-cases	HR (95% CI) ¹	p-val	N cases / non-cases	HR (95% CI) ¹	p-val	N cases / non-cases	HR (95% CI) ¹	p-val
	Excluding prevalent cases of obesity (N=94,366)			Excluding prevalent cases of diabetes (N=101,415)			Excluding prevalent cases of apnea (N=45,645)		
Overall cardiovascular diseases									
Time of first meal (1h incr.)	1,866 / 92,500	1.07 (1.01 – 1.13)	0.01	1,879 / 99,536	1.06 (1.00 – 1.11)	0.04	1,520 / 44,125	1.07 (1.01 – 1.14)	0.03
Time of last meal (1h incr.)	1,866 / 92,500	1.01 (0.97 – 1.06)	0.6	1,879 / 99,536	1.01 (0.97 – 1.06)	0.5	1,520 / 44,125	1.00 (0.95 – 1.06)	0.9
Number of eating occasions (1 occasion incr.)	1,866 / 92,500	0.99 (0.96 – 1.02)	0.4	1,879 / 99,536	0.99 (0.96 – 1.01)	0.3	1,520 / 44,125	0.99 (0.95 – 1.02)	0.5
Cerebrovascular diseases‡									
Time of first meal	902 / 93,464	1.06 (0.99 – 1.15)	0.1	938 / 100,477	1.04 (0.97 – 1.12)	0.3	752 / 44,893	1.07 (0.98 – 1.17)	0.1
Time of last meal	902 / 93,464	1.07 (1.00 – 1.15)	0.04	938 / 100,477	1.08 (1.01 – 1.15)	0.02	752 / 44,893	1.04 (0.96 – 1.12)	0.4
Number of eating occasions	902 / 93,464	0.97 (0.92 – 1.01)	0.1	938 / 100,477	0.96 (0.92 – 1.01)	0.09	752 / 44,893	0.97 (0.93 – 1.02)	0.3
Coronary heart diseases §									
Time of first meal	986 / 93,380	1.06 (0.98 – 1.14)	0.1	964 / 100,451	1.05 (0.98 – 1.13)	0.1	785 / 44,860	1.06 (0.97 – 1.15)	0.2
Time of last meal	986 / 93,380	0.97 (0.91 – 1.03)	0.3	964 / 100,451	0.96 (0.90 – 1.02)	0.2	785 / 44,860	0.97 (0.90 – 1.05)	0.5
Number of eating occasions	986 / 93,380	1.01 (0.97 – 1.05)	0.7	964 / 100,451	1.01 (0.97 – 1.05)	0.7	785 / 44,860	1.00 (0.96 – 1.05)	0.9

HR= Hazard ratio; N= Sample size; CI= Confidence Interval.

‡ Stroke and transient ischemic attack. § Myocardial infarction, acute coronary syndrome, angioplasty and angina pectoris.

1. Multivariable Cox proportional hazard models adjusted for age (timescale), sex (women, men), educational level (less than high school degree, <2 years after high school degree, ≥2 years after high school degree), monthly income per unit of consumption (less than 900€, 900-1,200€, 1,200-1,800€, 1,800-2,300€, 2,300-3,700€, more than 3,700€, don't want to answer), BMI at baseline (continuous, kg/m²), family history of CVDs (no, yes), alcohol consumption (Non-consumers (0g/day), low consumers (0.1 – 4.9 g/day), moderate consumers (5.0 – 14.9 g/day), high consumers (15.0 -29.9 g/day) and very high consumers (>30.0 g/day)), episodes of binge drinking (None, one, more than one), daily energy intake excluding alcohol (continuous, kcal/day), healthy and Western dietary patterns derived by factorial analysis (continuous), smoking (current regular (1 cigarette or more per day), current occasional, former, never), number of pack years (continuous, defined as the number of packs of cigarettes smoked per day by the number of years of smoking), physical activity (low, moderate, high) and number of dietary records (continuous). Time of first and last meal and number of eating occasions were mutually adjusted.

Reviewers' Comments:

Reviewer #1:

Remarks to the Author:

The authors seriously considered the feedback and they performed additional analyses, which strongly improved the manuscript. I still have some concerns that could not be resolved with these observational data, e.g. about nighttime snacking, influence of employment status, physical activity pattern, and the reason for waking up (natural vs. alarm etc.). Residual confounding and selection bias can therefore still be present and the data need to be interpreted with caution. Male participants (likely those at work) are underrepresented, which impacts the external validity of the findings. However, the authors now pay more attention to these limitations in the Discussion.

Reviewer #2:

Remarks to the Author:

The authors have done an excellent job addressing my comments.

REVIEWERS' COMMENTS

Reviewer #1 (Remarks to the Author):

The authors seriously considered the feedback and they performed additional analyses, which strongly improved the manuscript. I still have some concerns that could not be resolved with these observational data, e.g. about nighttime snacking, influence of employment status, physical activity pattern, and the reason for waking up (natural vs. alarm etc.). Residual confounding and selection bias can therefore still be present and the data need to be interpreted with caution. Male participants (likely those at work) are underrepresented, which impacts the external validity of the findings. However, the authors now pay more attention to these limitations in the Discussion.

We would like to thank Reviewer #1, for their thoughtful feedback and for acknowledging our efforts in response to the previous comments. We acknowledge that our study has limitations linked to its observational design. Following previous comments from Reviewer #1, we had conducted additional analyses and thoughtfully discussed in detail the potential residual confounding and selection bias in the limitations section. We have now highlighted these limitations again in the conclusion. Now it reads as follows (p. 19, l.584): “This work, which needs replication in other large-scale cohorts in different settings and using different and complementary approaches to minimize residual confounding and selection bias, supports an important role of adopting earlier eating timing patterns, consistently with previous experimental and observational studies. “

Reviewer #2 (Remarks to the Author):

The authors have done an excellent job addressing my comments.

We thank Reviewer #2 for the positive feedback. We are pleased to hear that you found our efforts in addressing your comments to be satisfactory.